# Technical Note: Characterising and comparing different palaeoclimates with dynamical systems theory

Gabriele Messori[1, 2] and Davide Faranda[3, 4, 5]

[1]Department of Earth Sciences, Uppsala University, and Centre of Natural Hazards and Disaster Science (CNDS), Uppsala, Sweden.
[2]Department of Meteorology and Bolin Centre for Climate Research, Stockholm University, Stockholm, Sweden
[3]Laboratoire des Sciences du Climat et de l'Environnement, LSCE/IPSL, CEA-CNRS-UVSQ, Université Paris-Saclay, Gif-sur-Yvette, France.
[4]London Mathematical Laboratory, London, U. K.
[5]LMD/IPSL, Ecole Normale Superieure, PSL research University, Paris, France

**Correspondence:** Gabriele Messori (gabriele.messori@geo.uu.se)

**Abstract.** Numerical climate simulations produce vast amounts of high-resolution data. This poses new challenges to the palaeoclimate community – and indeed to the broader climate community – in how to efficiently process and interpret model output. The palaeoclimate community also faces the additional challenge of having to characterise and compare a much broader range of climates than encountered in other subfields of climate science. Here we propose an analysis framework, grounded in dynamical systems theory, which may contribute to overcome these challenges. The framework enables to characterise the dynamics of a given climate through a small number of metrics. These may be applied to individual climate variables or to several variables at once, and diagnose properties such as persistence, active number of degrees of freedom and coupling. Crucially, the metrics provide information on instantaneous states of the chosen variable(s). To illustrate the framework's applicability, we analyse three numerical simulations of mid-Holocene climates over North Africa, under different boundary conditions. We find that the three simulations produce climate systems with different dynamical properties, such as persistence of the spatial precipitation patterns and coupling between precipitation and large-scale sea-level pressure patterns, which are reflected in the dynamical systems metrics. We conclude that the dynamical systems framework holds significant potential for analysing palaeoclimate simulations. At the same time, an appraisal of the framework's limitations suggests that it should be viewed as a complement to more conventional analyses, rather than as a wholesale substitute.

## 1 Motivation

Numerical climate models have enjoyed widespread use in palaeoclimate studies, from early investigations based on simple thermodynamic or general circulation models (e.g. Gates, 1976; Donn and Shaw, 1977; Barron et al., 1980) to the state-of-the-art models being used in the fourth phase of the Paleoclimate Modelling Intercomparison Project (PMIP4, Kageyama et al.,

2018). Compared to data from palaeo-archives, which is typically geographically sparse and with a low temporal resolution even for the more recent palaeoclimates (e.g. Bartlein et al., 2011), numerical climate simulations produce a vast amount of horizontally gridded, vertically resolved data with a high temporal resolution. Moreover, the resolution and complexity of numerical models – and hence the amount of data they produce – has increased vastly in recent years. This poses new challenges to the palaeoclimate community in how to efficiently process and interpret model output – indeed an issue which is

faced by the broader climate community (Schnase et al., 2016).

A related, yet distinct, challenge faced by the palaeoclimate community are the large uncertainties often found in palaeo-simulations. These reflect the uncertainties in palaeo-archives and in our knowledge of the boundary conditions and forcings affecting past climates (e.g. Kageyama et al., 2018). Thus, different simulations of the climate in the same period and region may yield very different results. This emerges in both reconstructions of climates from millions of years ago, such as the mid-

Pliocene warm period over 3 Myr BP (e.g. Haywood et al., 2013) and in climates much closer to us, such as the mid-Holocene around 6000 yr BP (e.g. Pausata et al., 2016). Characterising and understanding these discrepancies, requires analysis tools which may efficiently distil the differences between the simulated palaeoclimates.

Here, we propose an analysis framework which addresses the challenges of efficiently processing and interpreting large amounts of model output to compare different simulated palaeoclimates. The framework is grounded in dynamical systems

theory, and enables to characterise the dynamics of a given dynamical system – for example the atmosphere – through three one-dimensional metrics. A first metric estimates the persistence of instantaneous states of the system. We term this metric rather mundanely "persistence". A second metric, which we term "local dimension", provides information on how the system evolves to or from instantaneous states. Finally, the co-recurrence ratio is a metric applicable to two (or more) variables, which quantifies their instantaneous coupling. In other words, the dynamical information embedded in 3-dimensional (latitude,

longitude and time) or 4-dimensional (latitude, longitude, pressure level and time) data, commonly produced by climate models, can be projected onto three metrics which each provide a single value for every time-step in the data. These may then be interpreted and compared with relative ease.

The rest of this technical note is structured as follows: in Section 2 we briefly describe the theory underlying the dynamical systems framework, and provide both a qualitative and a technical description of the metrics. We further provide a link to a

repository from which MatLab code to implement the metrics may be obtained. In Section 3, we illustrate the application of the metrics to palaeoclimate data and their interpretation by using a set of recent numerical simulations for the mid-Holocene climate in North Africa. This is not meant to be a comprehensive analysis, but rather provide a flavour of the information provided by our framework. We conclude in Section 4 by reflecting on the framework's strengths and limitations and by outlining potential applications in future palaeoclimate studies.

## 2   A qualitative overview and theoretical underpinnings of the dynamical systems framework

In Sect. 2.1, we explain qualitatively how the dynamical systems metrics we use may be interpreted when computed for a hypothetical sea-level pressure (SLP) field. We also provide a conceptual analogy to raindrops flowing on complex topography.

In Sect. 2.2, we provide a brief mathematical derivation of the metrics, and outline some of the obstacles encountered when computing them for climate data.

## 2.1 A qualitative overview of the dynamical systems framework

The dynamical systems framework we propose rests on three indicators. All are instantaneous in time, meaning that given a long data series, they provide a value for each timestep. For example, if we were to analyse daily latitude-longitude SLP in a given geographical region over 30 years, we would have $30 \times 365$ values for each indicator.

The first indicator, termed *local dimension* ($d$), provides a proxy for the number of active degrees of freedom of the system about a state of interest (Lucarini et al., 2016; Faranda et al., 2017a). In other words, the value of $d$ for a given day in our SLP dataset tells us how the SLP in the chosen geographical region can evolve to or from the pattern it displays on that day. The number of different possible evolutions is proportional to the number of degrees of freedom, and therefore days with a low (high) local dimension correspond to SLP patterns that derive from and may evolve into a small (large) number of other SLP patterns in the preceding and following days. An intuitive – if not entirely precise – analogy may be drawn with the path followed by raindrops once they fall to the ground (Fig. A1a). In this case, an impermeable topography would effectively function as a bidimensional potential surface. If there is a deep valley, all raindrops falling on a small patch of ground on a side of the valley will follow similar paths: they will all reach the bottom of the valley and then flow along it. There may be places where the drops can follow slightly different paths, for example in going around a large stone, but their general course is constrained. This would be equivalent to a patch of ground with a low local dimension. Now imagine the case of raindrops falling on a second small patch of ground, but this time on a craggy mountain peak. In the latter case, very small changes in exactly where each drop falls may result in the drops following completely different paths. A first drop may follow a narrow crevasse to the side of the patch, while a second drop falling very close to it may take an entirely different route towards the bottom of the valley. This would be equivalent to having a high local dimension.

The second indicator, termed *persistence* ($\theta^{-1}$), measures the mean residence time of the system around a given state. In other words, if a given day in our SLP dataset has a high (low) persistence, the SLP pattern on that day has evolved slowly (rapidly) from and will evolve slowly (rapidly) to a different SLP pattern. The higher the persistence, the more likely it is that the SLP patterns on the days immediately preceding and following the chosen day will resemble the SLP pattern of that day. This metric is related to, yet distinct from, the notion of persistence issuing from weather regimes and similar partitionings of the atmospheric variability (Hochman et al., 2019, 2020b). Returning to our raindrop analogy (Fig. A1b), one may imagine that if the valley's sides are steep, the raindrops will leave the patch of ground they have fallen on very rapidly – namely, a low persistence. If, on the other hand, the valley is edged by shallow slopes, the raindrops will take longer to leave the patch – namely, a high persistence.

Both indicators may be used to characterise the dynamics underlying complex systems, including the Earth's climate (e.g. Faranda et al., 2017a; Buschow and Friederichs, 2018; Brunetti et al., 2019; Gualandi et al., 2020; Hochman et al., 2020a). On a more practical level, they can also be linked to the notion of intrinsic predictability of the system's different states. A state with a low $d$ and high $\theta^{-1}$ will afford a better predictability than one with a high $d$ and low $\theta^{-1}$. For more detailed discussions

on this topic, and a comparison to the conventional idea of predictability as evaluated through numerical weather forecasts, see Messori et al. (2017); Scher and Messori (2018) and Faranda et al. (2019a). Both $d$ and $\theta^{-1}$ may in principle be computed for more than one climate variable jointly (Faranda et al., 2020; De Luca et al., 2020b, a) but here we will focus on their univariate implementation.

Unlike the first two, the third metric we propose here, termed *co-recurrence ratio* ($\alpha$), is exclusively defined for two or more variables. Given two climate variables, $\alpha$ diagnoses the extent to which their recurrences co-occur, and hence provides a measure of the coupling between them (Faranda et al., 2020). As example, imagine that we now have both our SLP dataset and a corresponding precipitation dataset, which also provides daily values on a latitude-longitude grid. If $\alpha$ on a given day is large, then every time we have an SLP pattern on another day which closely resembles the SLP pattern of the chosen day (i.e. a *recurrence*), the precipitation pattern on that other day will also resemble the precipitation pattern of the chosen day. In other words, recurrences of similar SLP patterns lead to recurrences of similar precipitation patterns, which would suggest that the two variables are highly coupled. If $\alpha$ on a given day is small, then every time we have an SLP pattern on another day which closely resembles the SLP pattern of the chosen day, the precipitation pattern on that other day will not resemble the precipitation pattern of the chosen day. In other words, recurrences of similar SLP patterns do not lead to recurrences of similar precipitation patterns, which would suggest that the two variables are weakly coupled. We note that $\alpha$ may not be interpreted in terms of causation. However, since the joint recurrence of two fields implies the existence of a common underlying dynamics, the information it provides is nonetheless grounded in the physics of the system being analysed. Finally, $\alpha$ provides a very different information from many other conventional statistical dependence measures, since it gives a value for every timestep in the dataset. In our raindrop analogy, $\alpha$ could link the raindrop paths to, for example, the growth pattern of vegetation on the ground (Fig. A1c, d). Vegetation will affect the raindrop paths, yet the path of the raindrops will determine where the water flows, and hence affect the growth of the vegetation. One may therefore imagine that whenever the raindrops collectively follow similar paths, this will correspond to a recurring pattern of vegetation growth. Conversely, whenever there are similar patterns of vegetation growth, this will presumably result in the raindrops following similar paths. The raindrop paths and vegetation growth patterns therefore co-recur, resulting in a high $\alpha$.

## 2.2 Theoretical underpinnings of the dynamical systems framework

The three dynamical systems metrics described above, issue from the combination of extreme value theory with Poincaré recurrences (Freitas et al., 2010; Lucarini et al., 2012, 2016). We consider a stationary chaotic dynamical system possessing a compact attractor – namely the geometrical object hosting all possible states of the system. Given an infinitely long trajectory $x(t)$ describing the evolution of such system (in our previous example, our daily timeseries of SLP latitude-longitude maps) and a state of interest $\zeta_x$ (one specific SLP map), we define logarithmic returns as:

$$g(x(t),\zeta_x) = -\log[\text{dist}(x(t),\zeta_x)] \tag{1}$$

In this note, $dist$ is the Euclidean distance between two vectors. More generally, $dist$ can be a distance function which tends to zero as the two vectors increasingly resemble each other. For the implications of using $dist$ other than the Euclidean distance, the reader is referred to Lucarini et al. (2016) and Faranda et al. (2019b). The $-\log$ implies that $g(x(t), \zeta_x)$ attains large values when $x(t)$ and $\zeta_x$ are close to one another. We thus have a timeseries $g$ of logarithmic returns, which is large if $x$ at a specific time resembles the state of interest $\zeta_x$.

We next define a high threshold $s(q, \zeta_x)$ as the q$th$ quantile of $g(x(t), \zeta_x)$ (here $q = 0.98$), and define exceedances $u(\zeta_x) = g(x(t), \zeta_x) - s(q, \zeta_x) \ \forall \ g(x(t), \zeta_x) > s(q, \zeta_x)$. These are effectively the previously-mentioned Poincaré recurrences, for the chosen state $\zeta_x$. The interpretation of the above quantities for the idealised case of the Lorenz '63 attractor (Lorenz, 1963) – a simple three-dimensional dynamical system – is illustrated graphically in Fig. 1. We then leverage the Freitas-Freitas-Todd theorem (Freitas et al., 2010; Lucarini et al., 2012), which states that the cumulative probability distribution $F(u, \zeta)$ converges to the exponential member of the Generalised Pareto Distribution:

$$F(u, \zeta_x) \simeq \exp\left[-\vartheta(\zeta_x)\frac{u(\zeta_x)}{\sigma(\zeta_x)}\right] \tag{2}$$

Here, $u$ and $\sigma$ are parameters of the distribution which depend on the chosen $\zeta_x$, while $\vartheta$ is the extremal index: the standard measure of clustering in extreme value theory (Moloney et al., 2019). We estimate the latter using the Süveges Maximum Likelihood Estimator (Süveges, 2007). We then obtain the persistence as: $\theta^{-1}(\zeta_x) = \Delta t / \vartheta(\zeta_x)$, where $\Delta t$ is the timestep of the data being analysed, and the local dimension as $d(\zeta_x) = 1/\sigma(\zeta_x)$. The metrics' bounds are: $0 \leq d \leq +\infty$ and $0 \leq \theta \leq 1$.

Finally, we define the co-recurrence ratio by considering two trajectories $x(t)$ and $y(t)$, and a corresponding joint state of interest $\zeta = (\zeta_x, \zeta_y)$ (in our previous example, a specific SLP map and the associated precipitation map). We then have that:

$$\alpha(\zeta) = \frac{\nu[g(x(t)) > s_x(q) \mid g(y(t)) > s_y(q)]}{\nu[g(x(t)) > s_x(q)]} \tag{3}$$

with $0 \leq \alpha \leq 1$. Here, $\nu[-]$ is the number of events satisfying condition $[-]$, and all other variables are defined as before. By definition, $\alpha$ is symmetric with respect to the choice of variable ($x$ or $y$), since $\nu[g(x(t)) > s_x(q)] \equiv \nu[g(y(t)) > s_y(q)]$.

The above derivations all rely on the definition of recurrences relative to a threshold $s$. For cases where identical patterns repeatedly appear within the dataset being analysed, the threshold $s$ may become ill-defined – for example because the distance of the closest recurrences from the state of interest is 0 or because the distance of the closest recurrences exactly matches $s$. This does not imply that the dynamical system as a whole is repeatedly visiting identical states, but rather that the states are identical relative to the chosen variable – technically a so-called Poincaré Section of the full system. When one observes recurrences identical to the state of interest in the chosen variable, the asymptotic distribution of the exceedances $u(\zeta_x)$ is a Dirac delta. The state thus effectively has $d = 0$, namely the dimension of a point, and holds no dynamical information. In the current analysis, we chose to remove from our calculations both these $d = 0$ states and states whose closest recurrences exactly matched the relevant $s$. The upper bound in $d$ is numerically given by the size of the phase space being analysed – in our case by the number of gridpoints in the chosen geographical domain. The limiting case of $d = +\infty$ can be observed only for non-compact

attractors and thus does not apply to climate data. For persistence, $0 \leq \theta \leq 1$ implies that $1 \leq \theta^{-1} \leq +\infty$. $\theta$ can only be zero at a fixed point of the system, i.e. if all successive timesteps bring no change to the state of the system. A trivial example is a pendulum in its equilibrium position (or the equilibrium climate of a hypothetical planet with temperature 0 K and without any external energy input). The case $\theta = 1$ instead corresponds to non-persistent states of the dynamics, at least at the time resolution of the chosen data. The above-mentioned issue of an ill-defined $s$ also precludes the use of the Süveges estimator for $\vartheta$. In the case of identical patterns repeating within the dataset, the persistence $\theta^{-1}$ may be computed simply as the average number of consecutive identical timesteps in the variable being analysed. For consistency with the $d$ data, we however chose to exclude cases where the Süveges estimator could not be applied from our analysis. Finally, the lower ($\alpha = 0$) and upper ($\alpha = 1$) bounds of the co-recurrence ratio correspond to uncoupled dynamics and perfectly coupled dynamics, respectively. The above ill-defined threshold examples preclude computing $\alpha$, as $\nu[-] = 0$.

The analytical derivation of the above framework makes a number of assumptions which are typically not realised for climate data. For example, one has to take into account both the finite length of the datasets, and non-stationarities such as those issuing from internal low-frequency variability or varying external forcing. A formal justification of the applicability of the dynamical systems metrics to finite data issues from the results of Caby et al. (2020). There, the authors show that finite-time deviations of $d$ and $\theta$ from the asymptotic, unknown values contain information about the underlying system, since they are linked to the presence of unstable or periodic points of the dynamics. Similarly, both analytical and empirical evidence from Pons et al. (2020) shows that, although affected by the curse of dimensionality, estimates of $d$ from finite timeseries may be used in a relative sense to characterise the dynamics of a system – i.e. by comparing values of $d$ to one another. The conclusions drawn from these more theoretical results match those issuing from empirical tests on climate timeseries of finite length conducted by Buschow and Friederichs (2018). In practice, the two metrics may thus be applied to a variety of datasets issuing from chaotic dynamical systems, including (weakly non-stationary) climate datasets (e.g. Faranda et al., 2019c, 2020; Brunetti et al., 2019). MatLab code to compute $d$, $\theta^{-1}$ and $\alpha$ is provided at the end of this paper under "code availability".

## 3 Dynamical systems in action: an example from the mid-Holocene Green Sahara

### 3.1 The mid-Holocene Green Sahara: background and data

Today, the Sahara is the largest hot desert on Earth. Most of the precipitation in North-Western Africa is associated with the West African Monsoon (WAM), which reaches to around 16-17 °N (e.g. Sultan and Janicot, 2003) and effectively sets the boundary between the semiarid Sahel and the Sahara. However, the region has repeatedly experienced momentous hydroclimatic shifts in the past. In particular, there have been several periods when the Sahara was wetter and greener than today, often termed *African Humid Periods* (AHPs, see Claussen et al. (2017) and Pausata et al. (2020) for recent reviews on the topic).

The most recent AHP peaked during the mid-Holocene (MH), approximately 9000 yr – 6000 yr BP. It is thought to have coincided with an intensification and northward shift of the WAM, allowing the presence of vegetation, lakes and wetlands in areas that today are desert (e.g. Holmes, 2008, and references therein). Palaeo-archives suggest that during the MH AHP,

summer precipitation reached the northern parts of the present-day desert (e.g. Sha et al., 2019) and that tropical vegetation may have extended as far as 24 °N (Hély et al., 2014).

Numerical climate simulations of the MH have struggled to reproduce the full extent of the monsoonal intensification suggested by the palaeo-archives, and commonly suffer from a dry bias (Harrison et al., 2014). Early investigations on the topic highlighted the large sensitivity of the simulations to land-surface characteristics (e.g. Kutzbach et al., 1996; Kutzbach and Liu, 1997; Claussen and Gayler, 1997). More recent modelling efforts have confirmed this, and have further highlighted the potential role of an incorrect representation of atmospheric aerosols in favouring the dry bias (Pausata et al., 2016; Gaetani et al., 2017; Messori et al., 2019). Such hypothesis has triggered a lively discussion in the literature (cf. Thompson et al., 2019; Hopcroft and Valdes, 2019).

Here, we analyse the simulations used in Messori et al. (2019), performed with the EC-Earth Earth System Model v3.1 (Hazeleger et al., 2010). The atmospheric model has a T159 horizontal spectral resolution and 62 vertical levels. The ocean model has a nominal horizontal resolution of $1°$ and 46 vertical levels. In all simulations, the vegetation and aerosol concentrations are prescribed.

To illustrate the dynamical systems approach described in Sect. 2, we consider three different simulations. The first is a MH control simulation ($MH_{CNTL}$), which follows the PMIP3 protocol in imposing pre-industrial vegetation and atmospheric dust concentrations (Braconnot et al., 2011). The second is a Green Sahara simulation ($MH_{GS+PD}$), which imposes shrubland over the region $11—33$ °N and $15$ °W $-35$ °E. The third is a Green Sahara simulation that, in addition to the vegetation, also imposes a strongly reduced atmospheric dust loading ($MH_{GS+RD}$). Indeed, a greening of the Sahara would intuitively correspond to decreased dust emissions and hence to a lower atmospheric loading, as also supported by palaeo-archives (Demenocal et al., 2000; McGee et al., 2013) and modelling studies (Egerer et al., 2016).

We analyse 30 years of daily data of sea-level pressure (SLP), 500 hPa geopotential height (Z500) and precipitation frequency (prp) for each simulation. Precipitation frequency is constructed by assigning a value of 1 to grid points and time steps with non-zero precipitation and a value of 0 otherwise. This is preferable to using raw precipitation data for estimating the dynamical systems metrics (and $d$ in particular), as discussed further in Langousis et al. (2009) and Faranda et al. (2017a). As a technical consideration, we underline that the binary discretisation does not affect the spatio-temporal fractal nature of the precipitation field (Lovejoy and Schertzer, 1985; Brunsell, 2010). Nonetheless, it does make the distance $d$ a fundamentally different kind of random variable than for SLP and Z500, because its density consists of a finite number of point masses. The members of the extreme value distribution family, on the other hand, are continuous functions. Although there is no complete theoretical framework for the application of extreme value theory to recurrences of discrete fields, the analysis by Hitz (2016) supports the physical relevance of the results. Another issue with the precipitation data is that there can be repeated identical precipitation patterns (e.g. when there is no precipitation over the chosen domain). The implications of this for estimating the dynamical systems metrics are discussed in Sect. 2.2. We define the pre-monsoon season as March, April and May (MAM) and the monsoon season as June, July, August and September (JJAS). These definitions are based on the present-day WAM climatology. We use them in our analysis as reference periods when comparing the three numerical simulations described above. We quantify statistically significant differences when comparing median values of datasets using the Wilcoxon rank sum test

(Wilcoxon, 1945) at the 1% significance level. For the geographical precipitation anomalies, one-sided 5% significance bounds are determined using bootstrap resampling with 1000 iterations.

## 3.2 A dynamical systems view of the Mid-Holocene Green Sahara

The main interest in analysing the above simulations lies in understanding whether and why they reproduce different hydro-
220 climates over the Sahelian-Saharan region. Our aim in this Section is not to systematically investigate these two aspects, but rather to illustrate how the dynamical systems framework proposed here can be used to characterise the individual simulations and provide a concise overview of the differences between them. We argue that such an approach can provide a valuable complement to conventional analyses, and we relate our results to those obtained in earlier studies (e.g. Pausata et al., 2016; Gaetani et al., 2017; Messori et al., 2019).

A simple composite of JJAS average precipitation immediately highlights large differences in the precipitation regimes, with the $\text{MH}_{GS+PD}$ simulation showing a large northward shift and intensification of the monsoonal precipitation compared to $\text{MH}_{CNTL}$ (Fig. 2a, b) and the $\text{MH}_{GS+RD}$ simulation showing an additional, albeit smaller, precipitation increase (Fig. 2c, d). However, this time-mean picture hides a number of complex dynamical changes in the WAM, which we investigate using our dynamical systems framework. We focus on the Northern WAM region ($12.5 - 30\,^{\circ}\text{N}$, $10\,^{\circ}\text{W} - 20\,^{\circ}\text{E}$, black box in Fig. 2a).
This domain is chosen to reflect the region of seasonal monsoon rainfall which we expect to be most affected by the changes in land surface and atmospheric dust loading. Results for a more geographically extended domain are shown in Appendix A.

We begin by studying the seasonality of $d$ and $\theta^{-1}$ for precipitation data. In the $\text{MH}_{CNTL}$ (Fig. 3a, blue curve), the local dimension displays a marked interannual variability for any given calendar day, which we ascribe to the large variability in the monsoonal precipitation reproduced by the model (Fig. 3c, blue curve). The fact that the local dimension's variability
peaks in the pre-monsoon season, while that of precipitation itself peaks during the monsoon season, is likely related to the use of precipitation frequency which makes the local dimension more sensitive to changes in the timing of rain onset than to rain amount. This provides an insight into the potentially large onset variations within the same model simulation – an aspect which does not emerge from the variability of the zonally averaged precipitation climatology. The seasonal cycle of the local dimension displays two peaks, roughly matching the onset and withdrawal phases of the monsoon, somewhat lower
values during the height of the summertime monsoon and the lowest values during the dry season. Previous studies have noted how transition seasons can display an increase in the local dimension of atmospheric fields, because the atmosphere explores configurations belonging to more than one season (Faranda et al., 2017b). In more technical terms, this would reflect a saddle-like point of the atmospheric dynamics. We therefore interpret the two local maxima in $d$ as reflecting the northward shift and retreat of the monsoonal rainfall. The local dimension in the $\text{MH}_{GS+PD}$ and $\text{MH}_{GS+RD}$ simulations (red and orange curves,
respectively) presents a similar seasonal cycle, yet with the first local maximum shifted to earlier in the year, the second local maximum shifted to later in the year (Fig. 3b) and lower values throughout the monsoon season. Indeed, the medians of $d$ during the monsoon season in the $\text{MH}_{GS+PD}$ and $\text{MH}_{GS+RD}$ simulations are significantly different to that in the $\text{MH}_{CNTL}$ simulation. The shift of the local maxima points to a lengthening of the monsoon season, with an earlier rainfall onset and a later withdrawal. The timing of the first local maximum in $d$ indeed coincides with a rapid increase in the zonally averaged

precipitation at the southern edge of the domain in the $MH_{GS+PD}$ and $MH_{GS+RD}$ simulations (Fig. 3c). Such a lenghtening of the monsoonal period under a greening of the Sahara was previously noted in Pausata et al. (2016) by adopting a monsoon duration algorithm. The seasonal cycle of $\theta$ in $MH_{CNTL}$ (Fig. 3b, blue curve) displays a very different pattern. Low values (high persistence) occur during the monsoon season while higher values (lower persistence) occur during the dry season, albeit with a very large spread. This may reflect sporadic rainfall events at the edges of the domain outside of the monsoon season, with more persistent precipitation patterns during the monsoon season. The $MH_{GS+PD}$ and $MH_{GS+RD}$ simulations (red and orange curves, respectively) display a similar seasonality, albeit with a longer high-persistence monsoonal period, higher persistence values during the latter period, and a more marked difference in values between the monsoonal and dry phases. This chiefly results from lower values during the monsoonal period, likely reflecting a more geographically extensive and persistent precipitation regime. The median $\theta$ values during the monsoon period in the $MH_{GS+PD}$ and $MH_{GS+RD}$ simulations are significantly different from that of the $MH_{CNTL}$ simulation. One may hypothesise that the increased persistence underlies a decrease in importance of transient, mesoscale convective systems for driving the monsoonal precipitation, in favour of a regional re-organisation of precipitation into larger-scale persistent features. This would also explain the decrease in $d$ during the Monsoon season in the $MH_{GS+PD}$ and $MH_{GS+RD}$ simulations relative to the $MH_{CNTL}$ case. Gaetani et al. (2017) investigated the spectral properties and mesoscale motions of the monsoonal circulation, and concluded that the greening of the Sahara and dust reduction suppress African Easterly Waves and their role in triggering precipitation. This supports the hypothesis we formulated here on the basis of the 1-D metric $\theta$. The above qualitative considerations are mostly insensitive to the exact choice of geographical domain (cf. Figs. 3 and A2).

The seasonal variations in $d$ and $\theta$ can also be related to variations in the dynamical indicators on shorter timescales. The fact that a rapid increase in $d$ and a corresponding decrease in $\theta$ coincide with the northward progression of monsoonal rainfall indeed suggests that concurring high $d$ values and low $\theta$ values on daily timescales may correspond to specific spatial precipitation patterns. To verify this, we compute composite rainfall anomalies during JJAS on days with concurrent $d$ anomalies above the $70th$ percentile and $\theta$ anomalies below the $30th$ percentile of the respective JJAS distributions (Fig. 4). These relatively broad ranges are needed to ensure a good sample of dates, since here we are imposing a condition on each of the two metrics simultaneously. The anomalies are defined as deviations from a daily seasonal cycle. For example, the climatological value of a given variable in a given simulation for the 22nd July, is the mean of that variable across all 22nd July days in the simulation. Applying a smoothing to the climatology leads to very minor quantitative changes in our results (not shown). In $MH_{CNTL}$ (Fig. 4a), the anomalies are limited to the southern part of the domain, as the bulk of the Sahara receives little or no precipitation even at the peak of the monsoon (see Fig. 2a). The spatial pattern of the anomalies is wave-like, albeit with limited statistical significance, pointing to the fact that the dynamical systems metrics may reflect modulations in African Easterly Wave activity (see e.g. Fig. 8 in Gaetani et al. (2017) and the discussion above). The $MH_{GS+PD}$ and $MH_{GS+RD}$ simulations instead display clear and statistically significant anomaly dipoles, oriented in a predominantly meridional direction but with some zonal asymmetry. These correspond to a nortward shift of the monsoonal precipitation relative to the climatology (Fig. 4b and c, respectively). The dipoles span the whole domain, and display the largest anomaly values in the $MH_{GS+RD}$ simulation. This is

indeed the simulation showing the largest total rainfall, as well as the strongest northward shift of the monsoonal precipitation range (Fig. 2d). Very similar results are obtained if the same calculation is repeated over a larger domain (Fig. A3).

We next try to understand the physical processes underlying the differences in precipitation in the three simulations, by computing the co-recurrence ratio $\alpha$ between prp and SLP (Fig. 5a). In $MH_{CNTL}$ (blue line), as the monsoonal precipitation progresses northwards the coupling between the two variables increases, peaking in the middle of the monsoon season and waning thereafter. The dry season is characterised by overall low coupling values. In the $MH_{GS+PD}$ and $MH_{GS+RD}$ simulations (red and orange curves, respectively), $\alpha$ displays two local minima in the pre-monsoon season and in fall. During the northward progression of precipitation and the peak monsoonal phase, the values are mostly higher than for the $MH_{CNTL}$ simulation. Indeed, the median $\alpha$ values during the monsoon season of the $MH_{GS+PD}$ and $MH_{GS+RD}$ simulations are significantly different to that of the $MH_{CNTL}$ simulation. Both simulations also show higher $\alpha$ values than $MH_{CNTL}$ during the dry season, although these values are generally lower than in the monsoonal period. Similar results are found when extending the geographical domain (cf. Figs. 5 and A4), albeit with slightly higher coupling values for the extended domain during the dry season. These are likely associated to the presence of more abundant wintertime precipitation at the latter domain's southern boundary (Fig. A5). The stronger coupling in the $MH_{GS+PD}$ and $MH_{GS+RD}$ simulations compared to the $MH_{CNTL}$ during the pre-monsoon and monsoon seasons, points to the role of circulation anomalies – reflected in the SLP field – in favouring the northwards extension of the monsoonal precipitation. This was indeed noted in Pausata et al. (2016) by analysing changes in lower-level atmospheric thickness related to the Saharan Heat Low (see also Lavaysse et al., 2009). The higher $\alpha$ values during wintertime in the $MH_{GS+PD}$ and $MH_{GS+RD}$ simulations, may once again be related to the presence of limited amounts of winter precipitation in the domain while precipitation is almost entirely absent in the $MH_{CNTL}$ simulation (Fig. A5). A similar picture is found for the co-recurrence ratio between prp and Z500 (Figs. 5b, A4b), highlighting the robust nature of the increased coupling between precipitation and large-scale atmospheric circulation features in the $MH_{GS+PD}$ and $MH_{GS+RD}$ simulations.

As for $d$ and $\theta$ above, one may relate the seasonal variations in $\alpha$ to the daily anomalies associated with large or small values of the metric. We specifically consider precipitation, SLP and Z500 anomalies (computed as in Fig. 4) on JJAS days when $\alpha$ exceeds the 95*th* percentile of its anomaly distribution. These "strong coupling" days may be conceptualised as days on which recurrent spatial large-scale circulation anomalies favour recurrent spatial precipitation anomalies. In $MH_{CNTL}$, this takes the form of significantly increased precipitation across the southern portion of the domain, favoured by negative SLP anomalies to the North of the strongest precipitation anomalies (Fig. 6a) and positive Z500 anomalies to the North of the negative SLP core (cf. Figs. 6a and 7a). These are likely the footprint of a strengthened heat low (see e.g. Fig. 2b in Lavaysse et al. (2009)), which favours a northward progression of the monsoonal precipitation. As noted above, the signal being limited to the southern part of the domain is due to the $MH_{CNTL}$ simulation displaying little or no precipitation in the more northerly parts of the domain. The $MH_{GS+PD}$ simulation shows a statistically significantly, predominantly zonal dipole, with positive precipitation anomalies in the eastern part of the domain and negative anomalies further west (Figs. 6b, 7b). On strong coupling days, the large-scale circulation therefore favours an eastward extension of precipitation into a region that, even under a vegetated Sahara, receives little precipitation (see Fig. 2b). The SLP composite anomalies broadly match the ones of the $MH_{CNTL}$

simulation, while the Z500 anomalies are much larger in magnitude. The $MH_{GS+RD}$ simulation resembles the $MH_{GS+PD}$ simulation for the Z500 case, albeit with weaker geopotential height anomalies (Fig. 7c). An inverted precipitation dipole, with a significantly drier eastern part of the domain and a significantly wetter North-Western part, is instead seen for the SLP composite (Fig. 6c). Comparable results are found when extending the geographical domain, with some differences that we partly ascribe to the effect of $\alpha$ capturing some tropical precipitation patterns at the southern edge of the domain (cf. Figs. 6 and 7 with Figs. A6 and A7). A hypothesis to explain the differences between the $MH_{GS+PD}$ and $MH_{GS+RD}$ simulations is that, in the latter, enhanced deep convection triggered by large upward heat fluxes over the Sahara plays a larger role in shaping precipitation (Gaetani et al., 2017). This is in agreement with the increased amount of locally recycled moisture over the Sahara driven by dust reduction under a vegetated Sahara, noted by Messori et al. (2019).

The above results illustrate some of the strengths and limitations of the analysis framework we propose in this note, which we discuss further in Section 4 below. If applied in the context of a full-length research paper, some of the hypotheses expounded here could be verified through additional analyses. These could include, for example, the use of lower-level atmospheric thickness or other tailored indicators of heat low activity, of atmospheric radiative and heat fluxes, and of moist static energy as an indicator of convection.

## 4    An appraisal of the dynamical systems framework in a palaeoclimate context

Palaeoclimate simulations of the same period and region may yield very different results, the understanding of which requires analysis tools that may efficiently distil the discrepancies and point to possible underlying drivers. In this technical note, we have outlined an analysis framework which can efficiently compare the salient dynamical features of different simulated palaeoclimates. The framework is grounded in dynamical systems theory, and rests on computing three metrics: the local dimension $d$, the persistence $\theta^{-1}$ and the co-recurrence ratio $\alpha$. The first two metrics inform on the evolution of a system about a given state of interest – for example how the atmosphere evolves to or from a given large-scale configuration. The third metric describes the coupling between different variables.

From a theoretical standpoint, the dynamical systems framework presents a number of advantages over other statistical approaches for the analysis of large amounts of climate data such as clustering, Principal Component Analysis or Canonical Correlation Analysis. The first two are often used to define climate variability modes or weather regimes. The $d$ and $\theta$ metrics reflect the information captured by partitioning the atmospheric variability into specific regimes (e.g. Faranda et al., 2017a; Hochman et al., 2019), yet they also provide additional information on how the atmosphere evolves within and between the regimes. Canonical Correlation Analysis, which identifies maximum-correlation linear combinations of two variables, provides information which largely overlaps that given by $\alpha$ (De Luca et al., 2020b). However, the latter may be flexibly applied to multivariate cases beyond two variables, without the need for specific adaptations (such as partial CCA). Further, while statistical techniques can provide valuable information on the evolution of the climate system, the dynamical indicators we propose here are rooted in the system's underlying dynamics. In other words, their values are projections of mathematical properties of the underlying equations of the system, even when these are unknown. For example, a low local dimension does

not only point to a specific metastable state of the dynamics – as is the case for a conventionally defined weather regime – but also informs that this state is in a predictable region of the attractor. Moreover, the computation of the dynamical systems metrics requires essentially a single free parameter to be fixed, namely the threshold to define recurrences, and one may easily
test the stability of the estimates with respect to small perturbations to this threshold. Furthermore, states with $\theta \to 0$ indicate quasi-singularities – technically unstable fixed points – of the system. Quasi-singular states portend tipping points or tipping elements of the climate system that have not yet been crossed (e.g. Lenton et al., 2008), and can thus be of interest for a range of palaeoclimate applications. Indeed, analysis of data issuing from both conceptual (Faranda et al., 2019c) and reduced-complexity (Messori et al., 2020) models of specific features of atmospheric dynamics have highlighted that changes in $d$ and
$\theta$ values reflect transitions between different basins of attraction of the system. Finally, the metrics provide one value for every timestep in the analysed data, and may be conveniently used to investigate seasonality, oscillatory behaviours, high-frequency variability and more. This is especially valuable for the co-recurrence ratio, as a number of other measures of coupling or correlation between two variables only provide a single value for the whole time period being considered.

Because of these characteristics, the dynamical systems metrics can be particularly helpful when processing large datasets
(see e.g. Rodrigues et al. (2018); Faranda et al. (2019a)). To illustrate their practical applicability in palaeoclimate studies, we have analysed three numerical simulations of the mid-Holocene climate over North Africa: a control simulation with pre-industrial vegetation and atmospheric dust loading, a Green Sahara simulation with shrubland imposed over a broad swath of what is today the Sahara desert, and a second Green Sahara simulation which additionally features heavily reduced atmospheric dust loading. Our aim is to show that the different hydroclimates in these simulations correspond to different dynamical
properties of the modelled climate systems, which are captured by the three dynamical systems metrics. The seasonal cycles of $d$ and $\theta^{-1}$ reflect features of the duration, interannual variability and geographical extent of the monsoon, which do not always emerge clearly from the precipitation's seasonal cycle. The metrics further capture the differences between the simulations, and may be leveraged to formulate hypotheses on their physical drivers, such as modulations in atmospheric wave activity. The co-recurrence ratio $\alpha$, which provides a temporally-resolved measure of coupling between different variables, enriches the
picture by enabling to contextualise precipitation changes relative to large-scale atmospheric circulation anomalies.

As a caveat, we note that our approach is more successful in providing insights on the changes between the control and each of the Green Sahara simulations, than between the latter two simulations. Previous analyses of these same simulations and studies from other authors (e.g. Pausata et al., 2016; Thompson et al., 2019) suggest that, compared to the effect of Saharan Greening, the dust reduction under a Green Sahara scenario only has limited impacts on the atmospheric circulation. This
points to our framework being best-suited for diagnosing shifts in palaeoclimate dynamics, as opposed to smaller climatological changes not associated to changes in the underlying driving processes.

Additionally, obtaining good estimates of $d$, $\theta^{-1}$ and $\alpha$ requires relatively long time series, limiting their applicability to palaeo-archives. At the same time, empirical evidence shows that daily timeseries of a few decades – as often obtained from numerical simulations – are typically sufficient for many climate applications. Indeed, previous studies have outlined that the
estimates of the metrics for atmospheric observables convergence relatively fast (e.g. Faranda et al., 2017a; Buschow and Friederichs, 2018). There are no fixed rules for determining the minimum required amount of data, but current best practice is

to have several good recurrences of the patterns of interest in the data. While no formal definition of what constitutes a "good recurrence" is forthcoming, a simple test that may be applied is to reduce the length of the datasets being used and repeat the metrics' estimates to check their stability (e.g. Buschow and Friederichs, 2018). Non-stationary data, such as may be found in
transient palaeoclimate simulations, also require some care in verifying that recurrences can be identified (see also Sect. 2). Our methodology is able to detect weak non-stationarities in the climate system, as for example is the case for the ongoing climate change (e.g. Faranda et al., 2019a). However, an abrupt regime shift poses a different challenge, and it is an open question as to what is the limit of validity of our metrics for non-stationary systems. A further difficulty that may be encountered in applying the dynamical systems framework pertains its interpretation. While the three metrics lend themselves to making relatively
intuitive heuristic inferences, they may sometimes provide counterintuitive results, such as Figs. 6c and 7c here, and there is no universally valid approach to overcome these interpretative difficulties. Furthermore, expounding formal arguments to support the results obtained requires a detailed knowledge of the underlying theoretical bases, which may initially be daunting.

In this technical note, we aimed to give a flavour of the dynamical systems framework's possible application to palaeoclimate simulations, as opposed to presenting a systematic analysis. We specifically wished to highlight its potential for comparing
different palaeoclimates, while also providing an appraisal of its limitations. To do so, we focussed on three existing simulations and on a small number of atmospheric variables. However, the approach may be applied to a very broad range of palaeoclimate applications, not limited to the comparison of different climates or to the atmosphere. In particular, the co-recurrence coefficient could be used to study interactions between the different components of the climate system varying on different timescales, such as the hydrosphere and the atmosphere, or the hydrosphere and the cryosphere (e.g. by comparing the response of different
numerical models to the same forcing). As mentioned above, $\theta$ may also have a direct application in the detection of tipping points or states. From a technical perspective, we envisage that the most effective application of the framework would be for the analysis of very large datasets, such as those issuing from the PMIP initiative or from downscaling efforts on very long transient simulations (e.g. Lorenz et al., 2016). At the same time, we stress that we do not view the framework as a wholesale substitute for conventional analyses of palaeoclimate dynamics. Rather, it is intended as a complement that may
help to strengthen mechanistic interpretations and rapidly identify features deserving further investigation.

*Code availability.* The code to compute the three dynamical systems indicators used in this study is made freely available through the cloud storage of the *Centre National de la Recherche Scientifique* (CNRS), under a CC BY-NC 3.0 license:

https://mycore.core-cloud.net/index.php/s/pLJw5XSYhe2ZmnZ

*Author contributions.* G. Messori conceived the study and performed the analysis. D. Faranda provided the publicly available code. Both
authors contributed to drafting the manuscript.

*Competing interests.* The authors have no competing interests to declare.

*Acknowledgements.* The authors thank F. Pausata, Q. Zhang and M. Gaetani for making the palaeoclimate simulations available. We also thank the two anonymous Reviewers for the detailed and pertinent comments they provided. G. Messori was partly supported by the Swedish Research Council Vetenskapsrådet (grant no. 2016-03724) and the Swedish Research Council FORMAS (grant no. 2018-00968). DF is
supported by a CNRS-INSU LEFE/MANU grant (DINCLIC project).

## Appendix A:  Additional Figures

In this appendix, we provide a schematic of the raindrop analogy for the dynamical systems metrics and figures illustrating the sensitivity of our results to the choice of geographical domain and season. The figures are discussed in the main text.

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

540

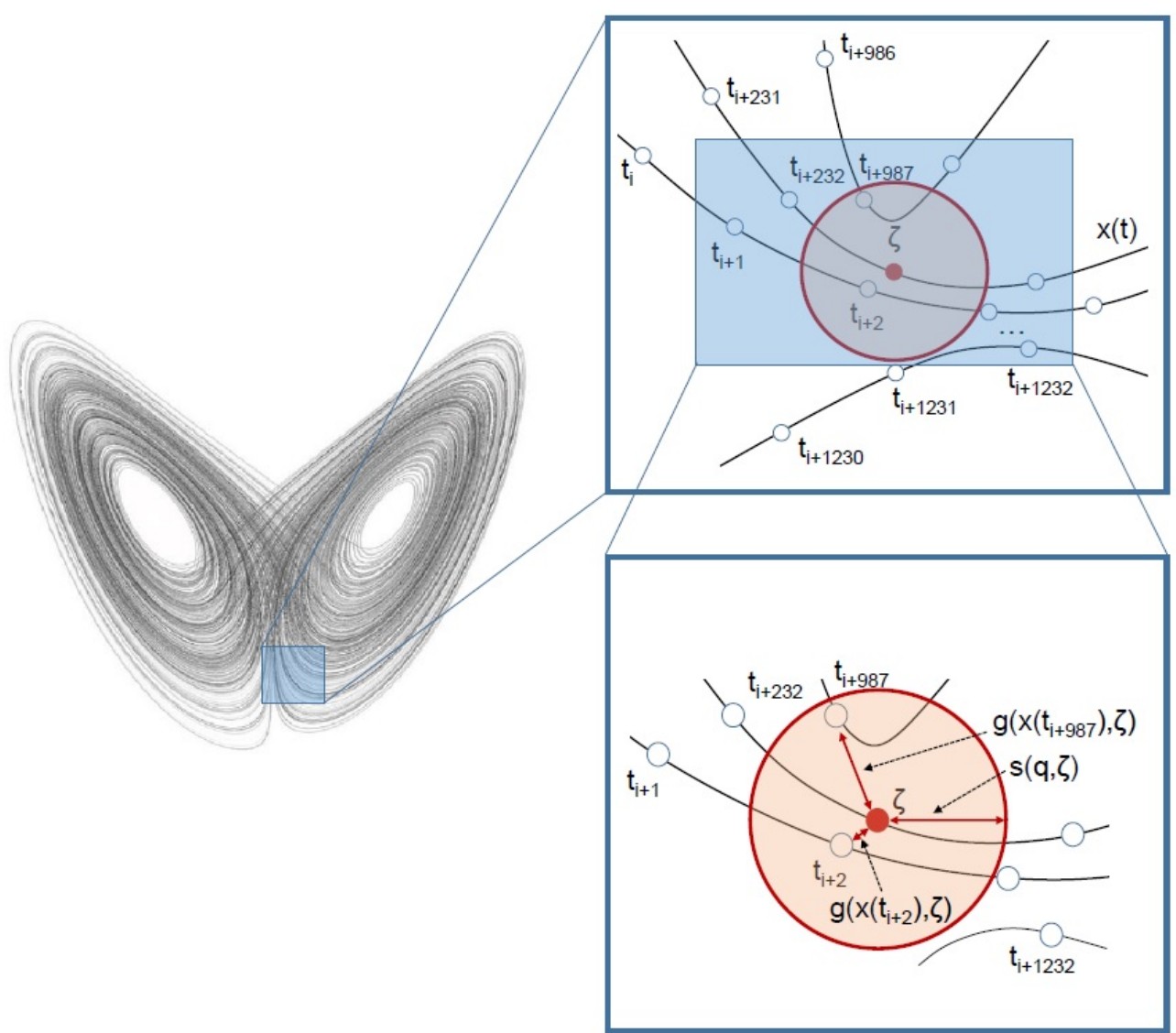

**Figure 1.** Schematic of the computation of the dynamical systems metrics for a state $\zeta$ on the Lorenz '63 attractor. All trajectory segments shown in the right hand side panels are part of a single, long trajectory $x(t)$. The white circles along the trajectory represent discrete, instantaneous measurements of the continuous evolution of the trajectory. The state of interest $\zeta$ is shown in red. The bottom right panel illustrates the hyper-sphere determined by the high threshold $s(q,\zeta)$, which defines recurrences, and the logarithmic distances between measurements defined by $g(x(t),\zeta)$. Here, $g$ takes large values for small separations. Thus, for all points within the hyper-sphere we have that $g(x(t),\zeta) > s(q,\zeta)$. In the schematic, only two measurements satisfy this condition (adapted from Faranda et al. (2020)).

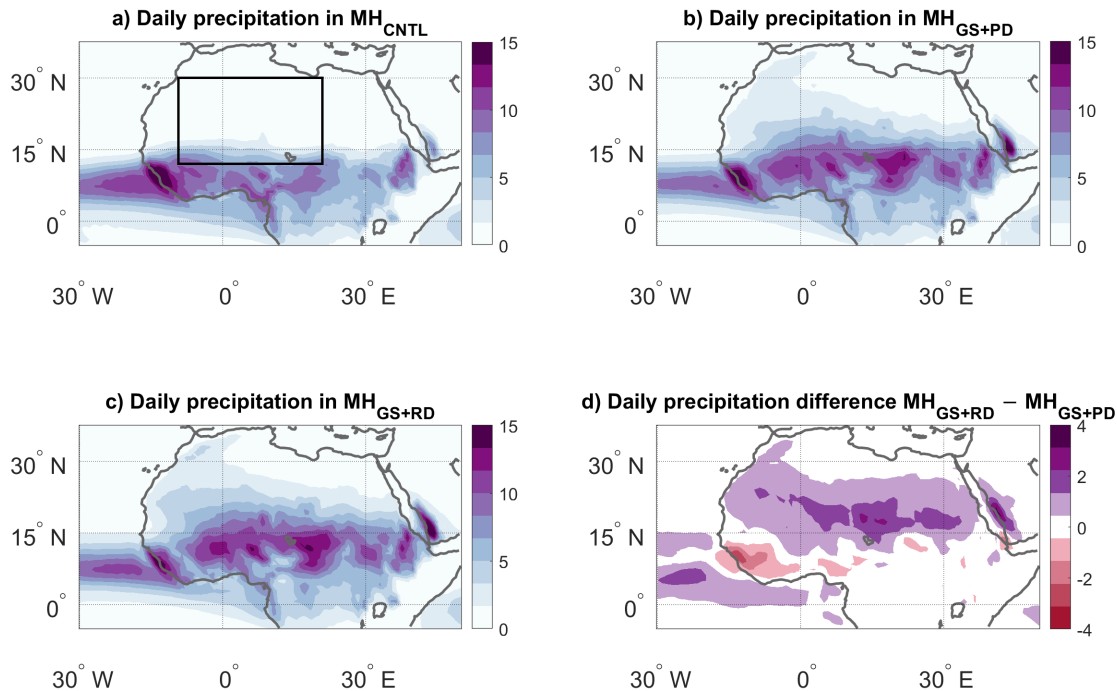

**Figure 2.** JJAS precipitation (mm day$^{-1}$) for the: (a) MH$_{CNTL}$, (b) MH$_{GS+PD}$ and (c) MH$_{GS+RD}$ simulations. (d) Precipitation difference between the MH$_{GS+RD}$ and MH$_{GS+PD}$ simulations. The black box in (a) marks the domain used to perform the dynamical systems analysis (12.5 − 30 °N, 10 °W − 20 °E).

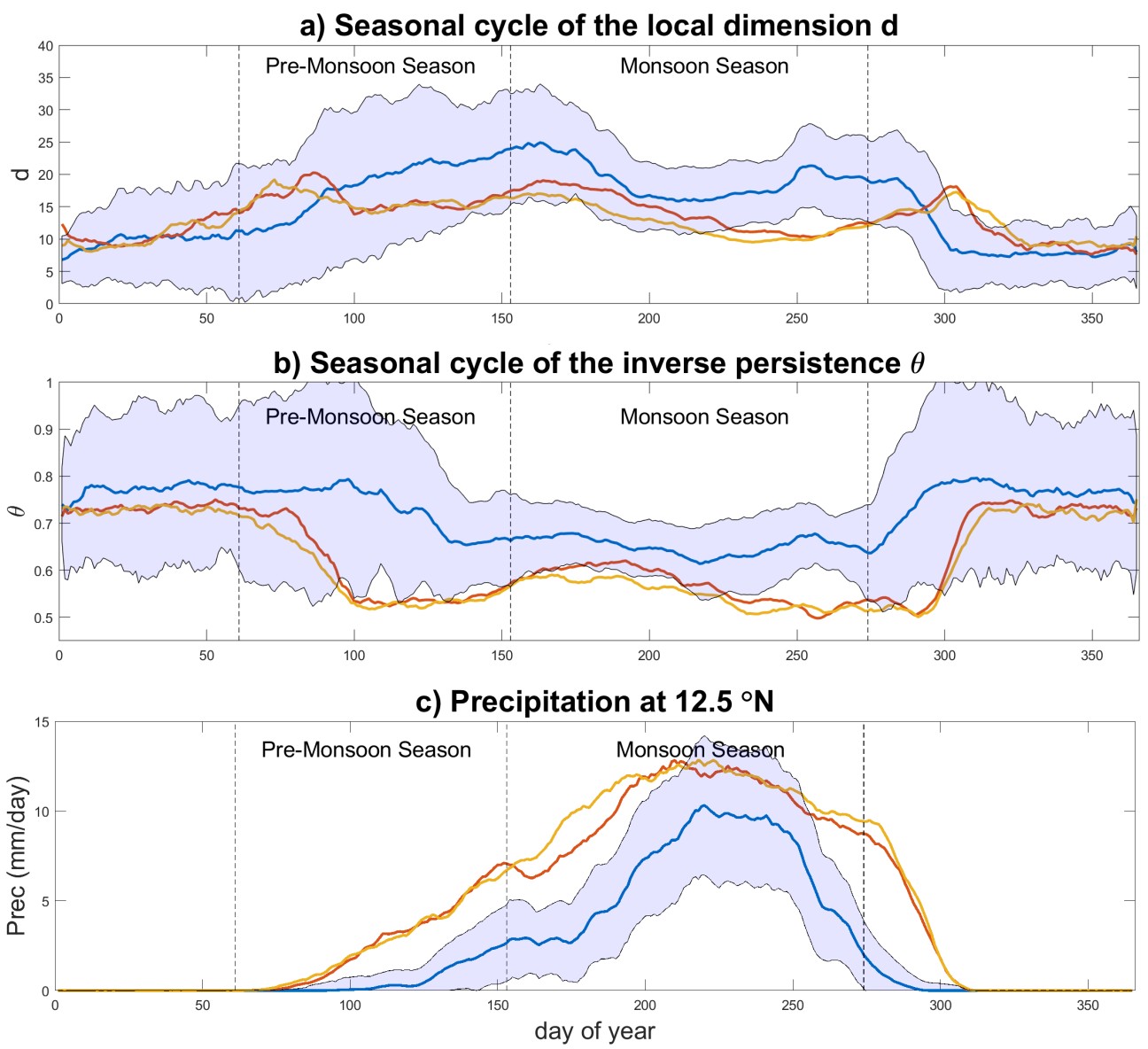

**Figure 3.** Seasonal cycle of median (a) $d$, (b) $\theta$ and (c) zonally averaged daily precipitation at 12.5 °N for the MH$_{CNTL}$ (blue), MH$_{GS+PD}$ (red) and MH$_{GS+RD}$ (orange) simulations. The blue shading marks $\pm$ 1 std from the MH$_{CNTL}$. The vertical dashed lines mark the pre-monsoon (MAM) and monsoon (JJAS) seasons. The data is smoothed with a 10-day moving average.

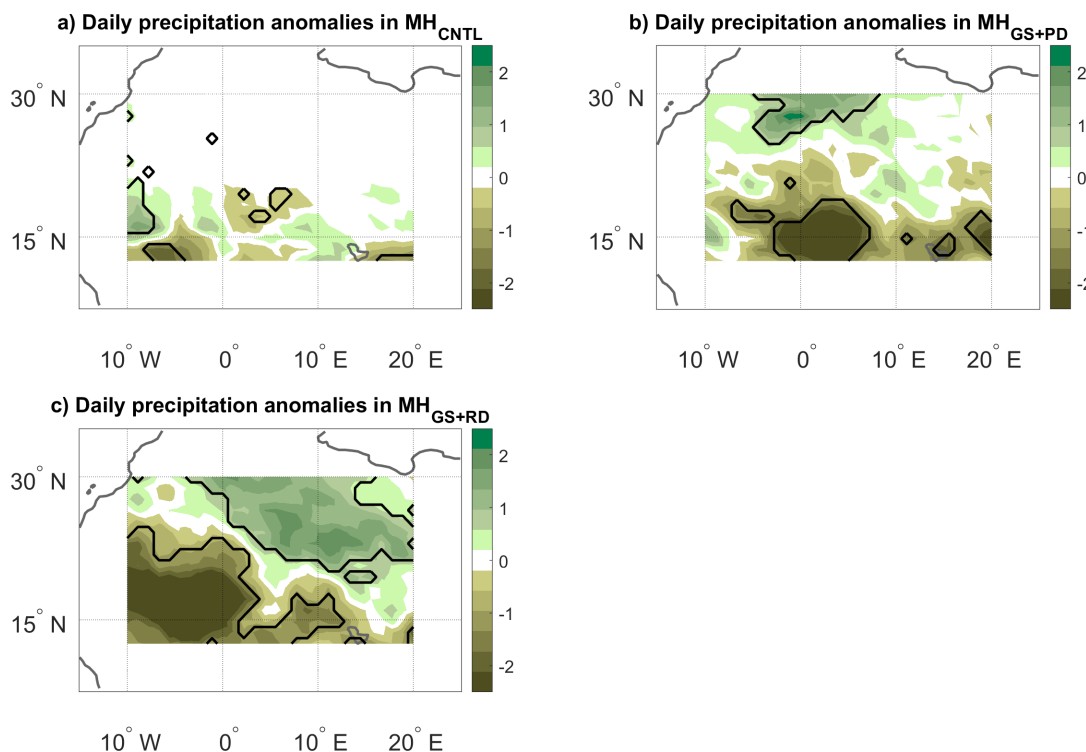

**Figure 4.** JJAS precipitation anomalies (mm day$^{-1}$) on days with high $d$ and low $\theta$ (see text) for the: (a) MH$_{CNTL}$, (b) MH$_{GS+PD}$ and (c) MH$_{GS+RD}$ simulations. The anomalies are only shown over the domain used to perform the dynamical systems analysis (see black box in Fig. 2a). Bold lines mark significance bounds (see text).

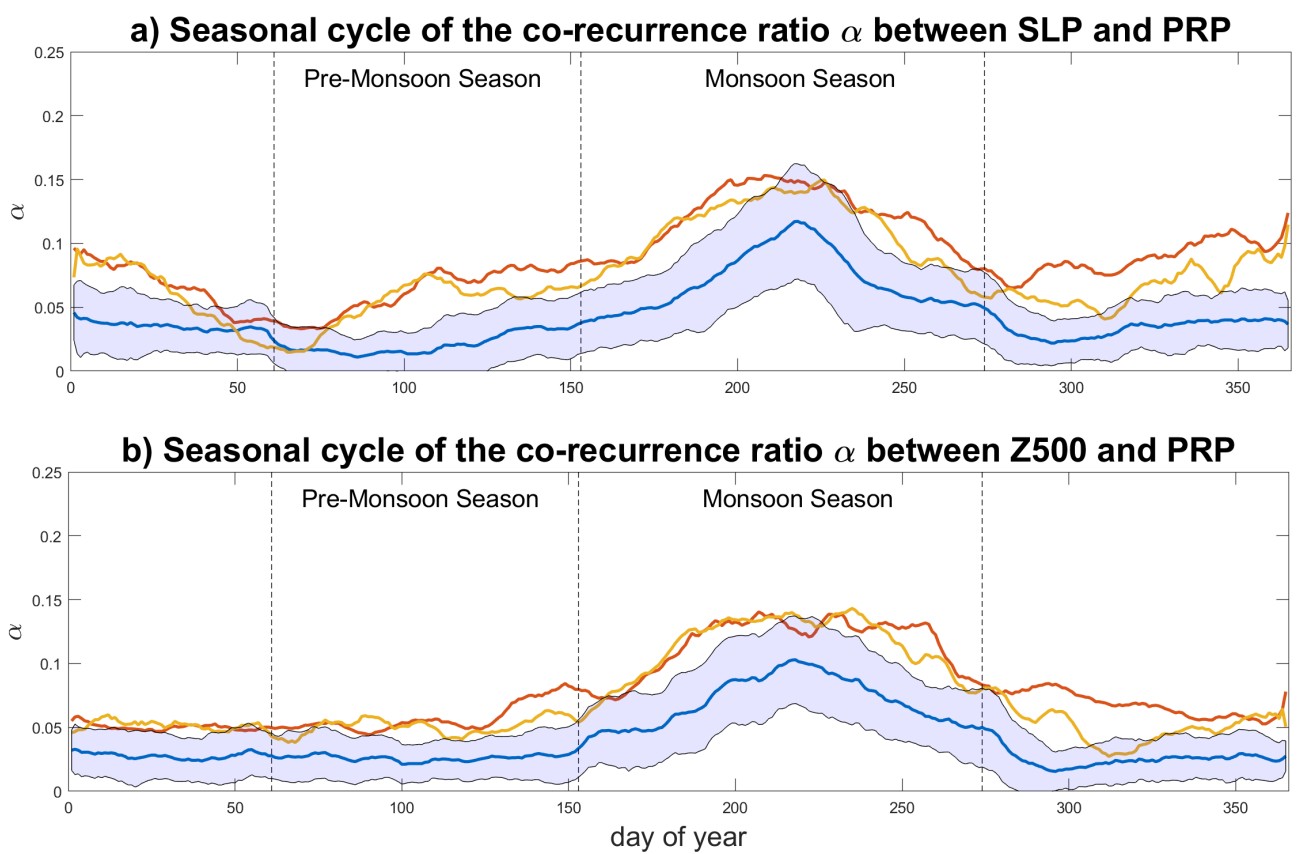

**Figure 5.** Seasonal cycle of median (a) $\alpha_{SLP,PRP}$ and (b) $\alpha_{Z500,PRP}$ for the MH$_{CNTL}$ (blue), MH$_{GS+PD}$ (red) and MH$_{GS+RD}$ (orange) simulations. The blue shading marks $\pm$ 1 std from the MH$_{CNTL}$. The vertical dashed lines mark the pre-monsoon (MAM) and monsoon (JJAS) seasons. The data is smoothed with a 10-day moving average.

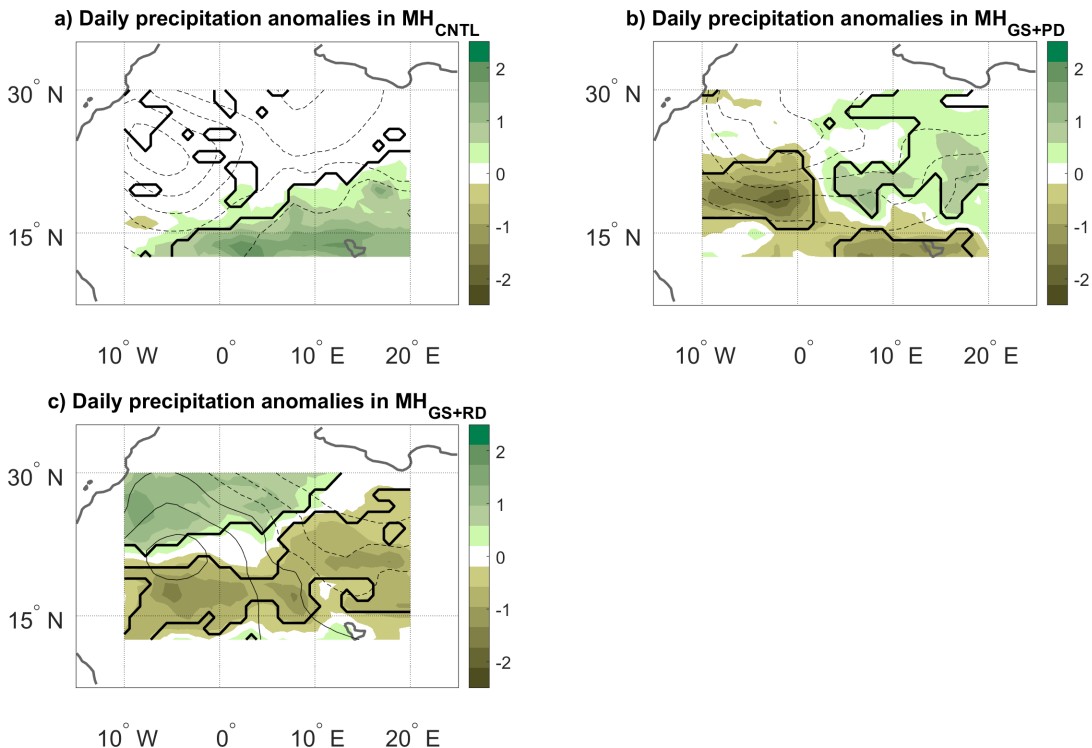

**Figure 6.** JJAS precipitation (colours, mm day$^{-1}$) and SLP (contours, hPa) anomalies on days with high $\alpha$ (see text) for the: (a) MH$_{CNTL}$, (b) MH$_{GS+PD}$ and (c) MH$_{GS+RD}$ simulations. The contour lines have an interval of 0.25 hPa and span the ranges: (a) -0.25 hPa to -1.5 hPa; (b) -0.25 hPa to -1.25 hPa; and (c) +0.5 hPa to -0.75 hPa. Continuous contours show zero and positive anomalies, dashed contours show negative anomalies. The anomalies are only shown over the domain used to perform the dynamical systems analysis (see black box in Fig. 2a). Bold lines mark significance bounds for precipitation (see text).

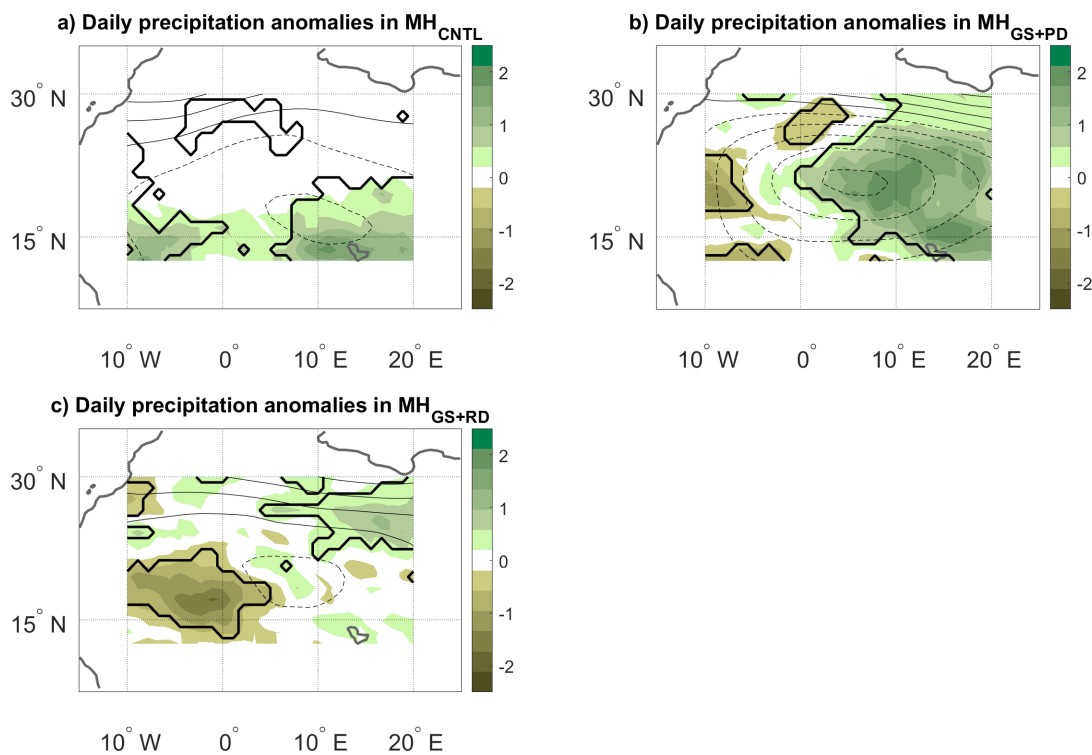

**Figure 7.** JJAS precipitation (colours, mm day$^{-1}$) and Z500 (contours, m) anomalies on days with high $\alpha$ (see text) for the: (a) MH$_{CNTL}$, (b) MH$_{GS+PD}$ and (c) MH$_{GS+RD}$ simulations. The contour lines have an interval of 20 m and span the ranges: (a) +40 m to -40 m; (b) +60 m to -100 m; and (c) +60 m to -20 m. Continuous contours show zero and positive anomalies, dashed contours show negative anomalies. The anomalies are only shown over the domain used to perform the dynamical systems analysis (see black box in Fig. 2a). Bold lines mark significance bounds for precipitation (see text).

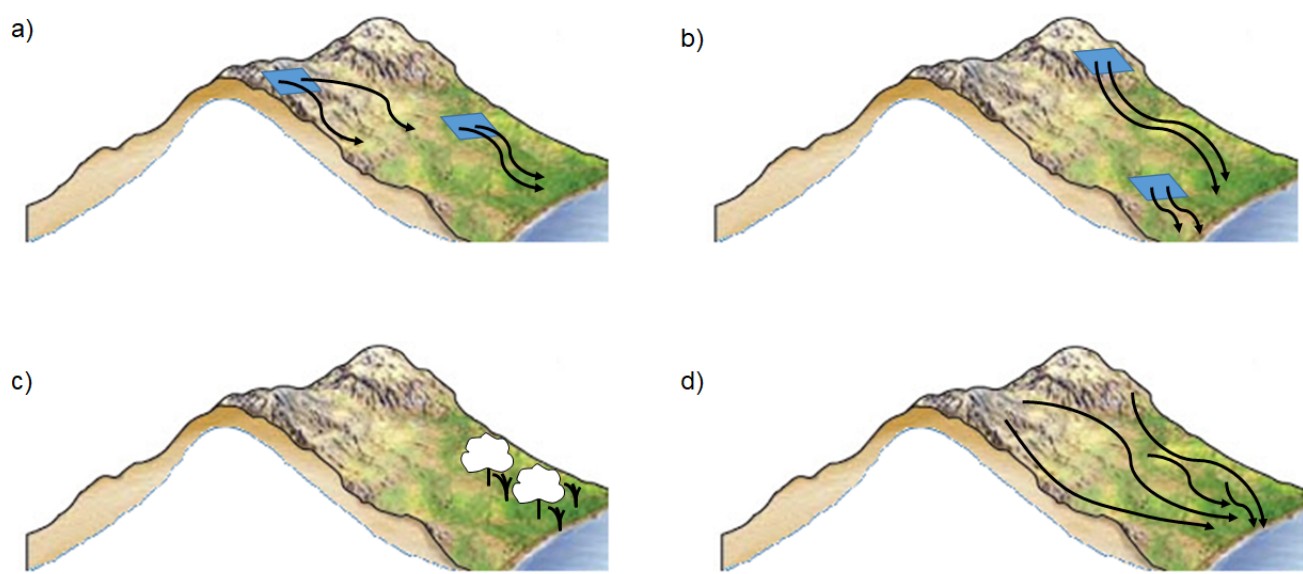

**Figure A1.** The raindrop analogy for the dynamical systems metrics. (a) Depending on the topography, raindrops falling on a small patch of ground on the side of a valley may follow similar paths (low local dimension) or different paths (high local dimension). (b) If the patch of ground is on a steep incline, the raindrops will leave it very rapidly (low persistence); if the incline is shallow, the raindrops will take longer to leave the patch (high persistence). (c, d) Vegetation will affect the raindrop paths, and at the same time the path of the raindrops will affect the growth of the vegetation. Whenever the raindrops collectively follow similar paths, this will correspond to a recurring pattern of vegetation growth, and vice-versa (high co-recurrence ratio). Part of the figure has been excerpted from Marshak (2019), with permission of the publisher, W. W. Norton & Company, Inc. All rights reserved (Copyright (c) 2019, 2015, 2012, 2008, 2005, 2001 by W. W. Norton & Company, Inc.).

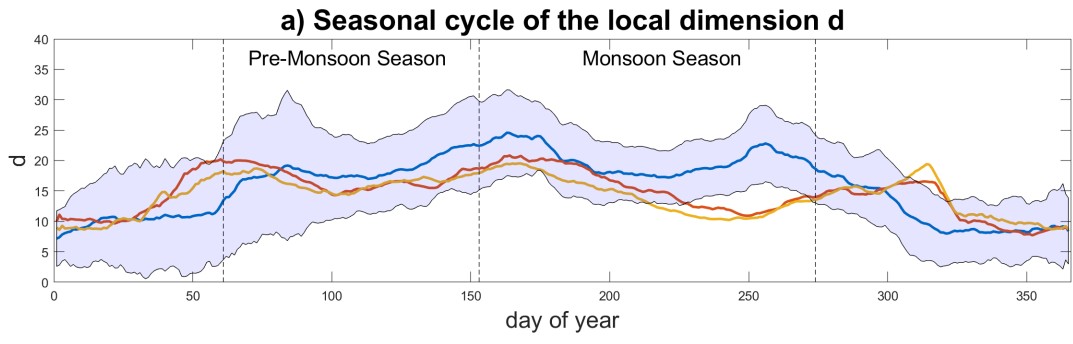

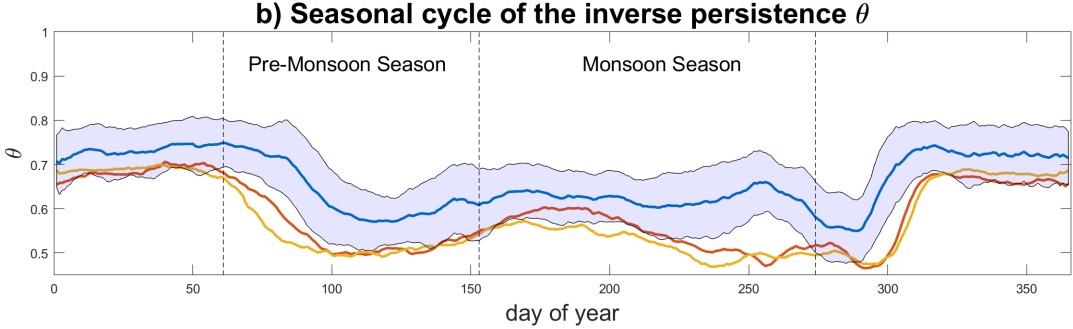

**Figure A2.** Same as Fig. 3a, b, but using the domain (10 – 30 °N, 15 °W – 20 °E).

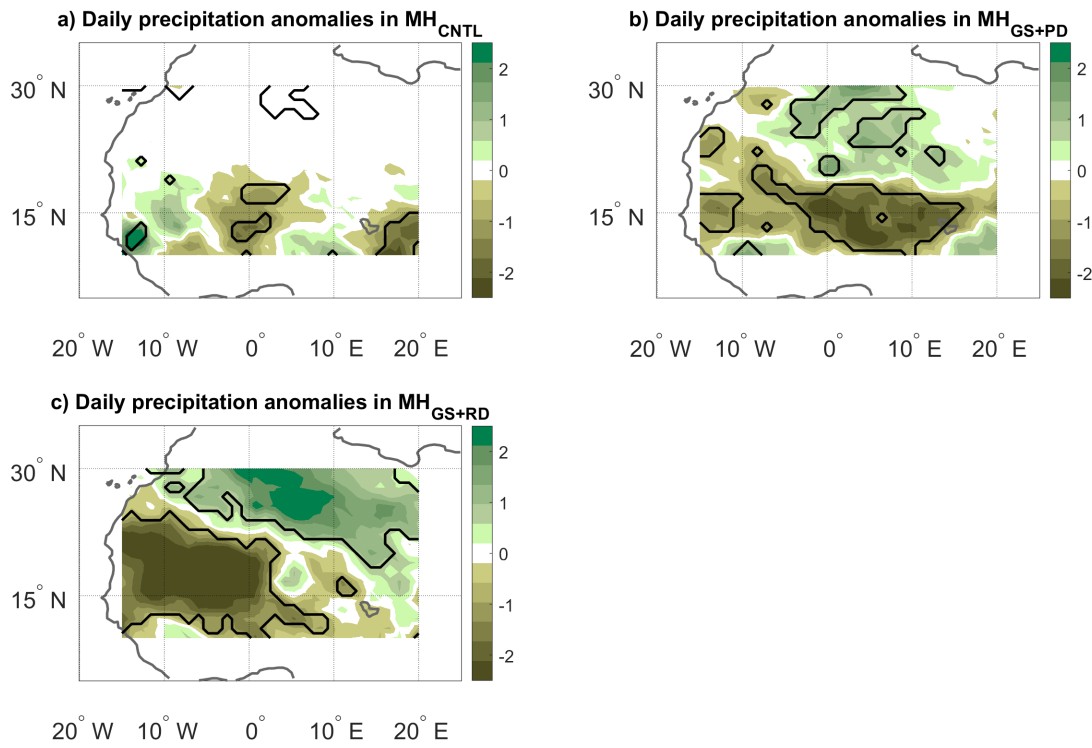

**Figure A3.** Same as Fig. 4, but using the domain (10 – 30 °N, 15 °W – 20 °E).

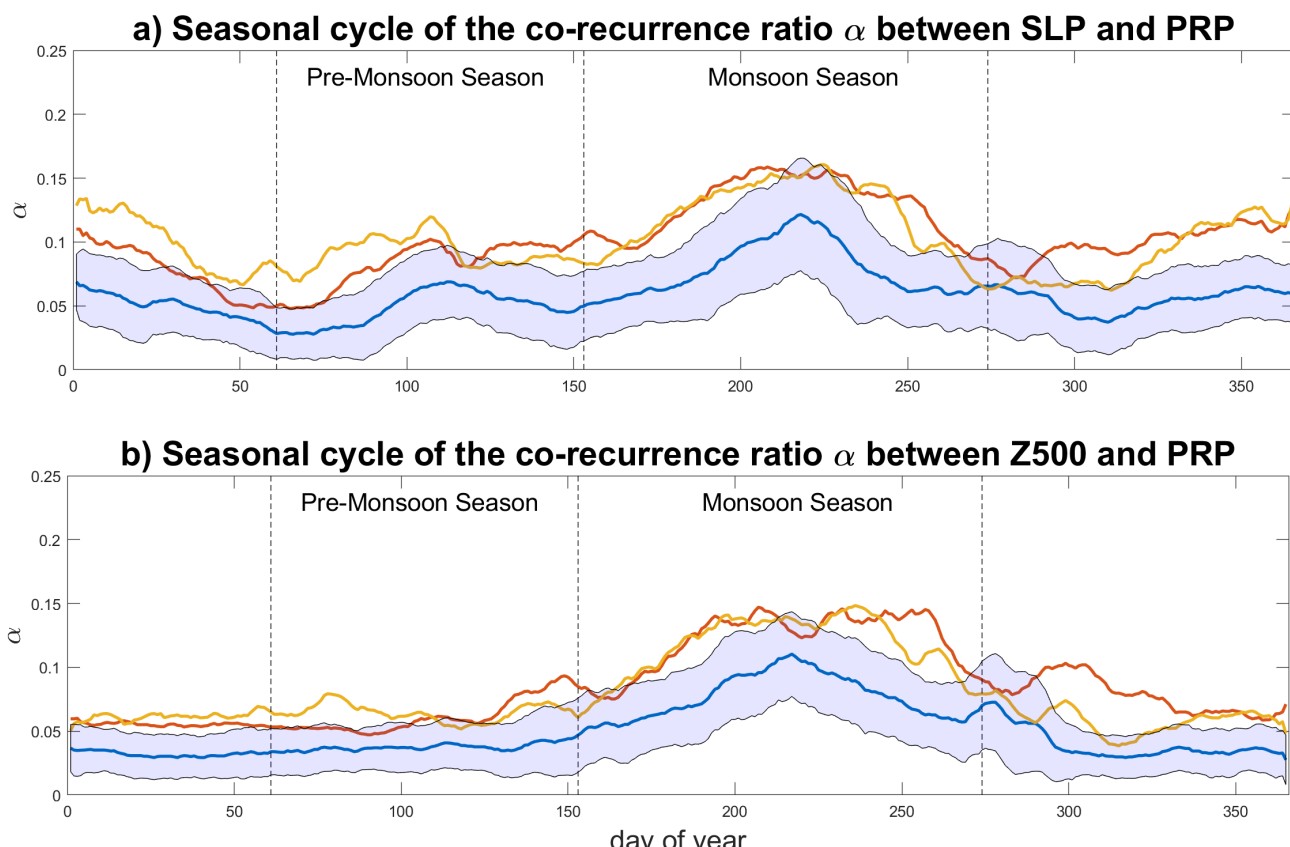

**Figure A4.** Same as Fig. 5, but using the domain (10 – 30 °N, 15 °W – 20 °E).

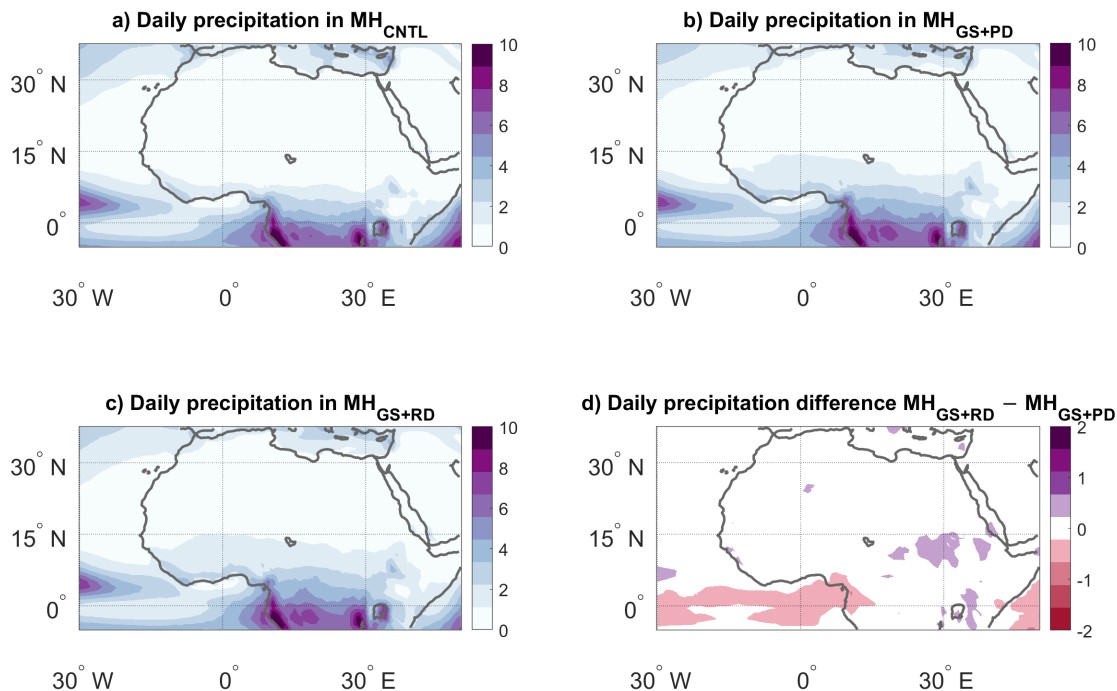

**Figure A5.** Same as Fig. 2, but for the October–February period.

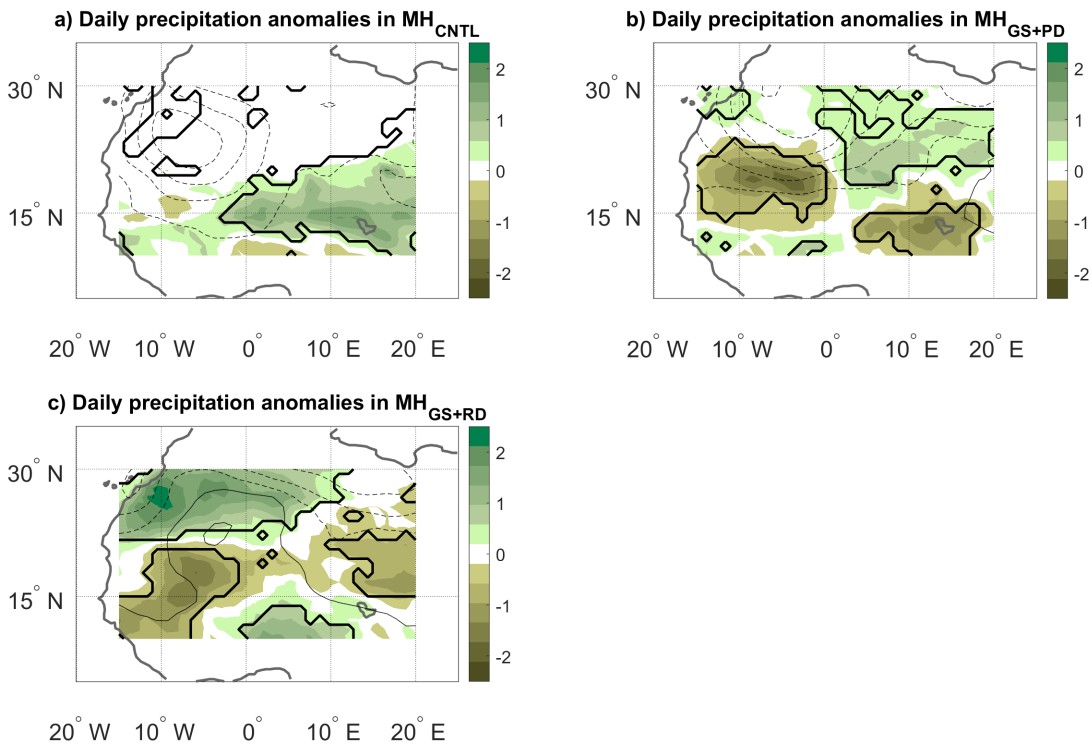

**Figure A6.** Same as Fig. 6, but using the domain (10 – 30 °N, 15 °W – 20 °E). The contour lines have an interval of 0.25 hPa m and span the ranges: (a) -0.25 hPa to -1.25 hPa; (b) 0 hPa to -1.5 hPa; and (c) +0.25 hPa to -0.75 hPa.

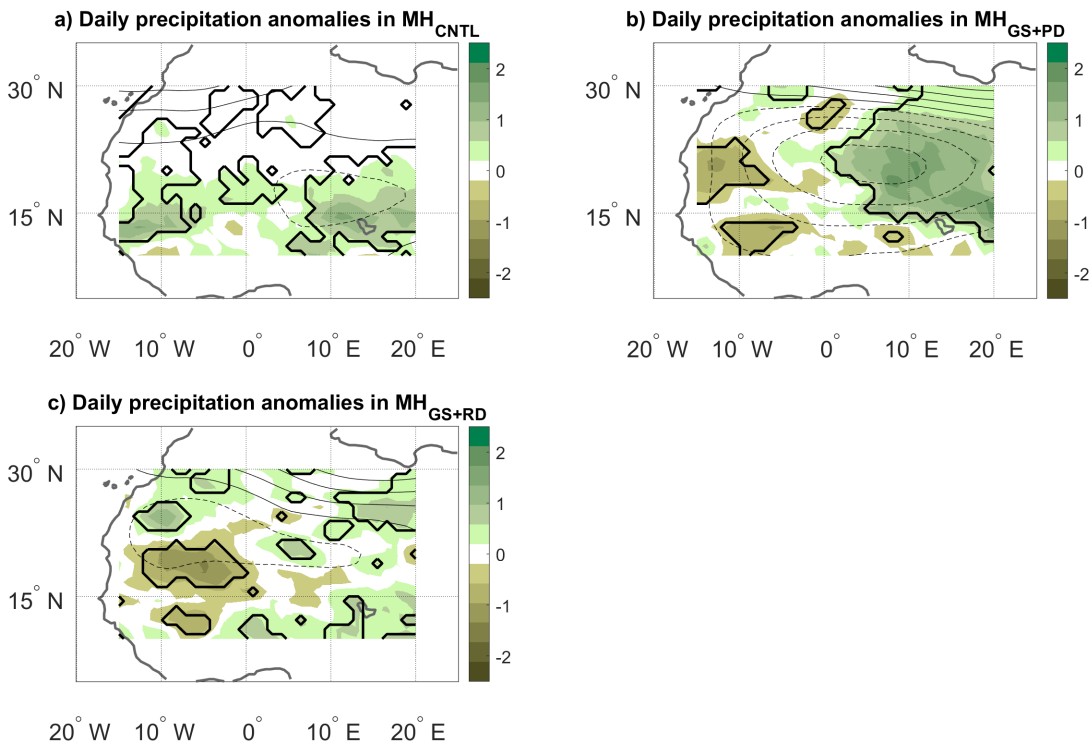

**Figure A7.** Same as Fig. 7, but using the domain (10 – 30 °N, 15 °W – 20 °E). The contour lines have an interval of 20 m and span the ranges: (a) +40 m to -20 m; (b) +60 m to -80 m; and (c) +60 m to -20 m.