# Peer review of "Technical Note: Characterising and comparing different palaeoclimates with dynamical systems theory"

_Climate of the Past, 2020_

## Referee Comment (RC1) · Anonymous Referee #1 · 13 Sep 2020

Review of "Technical Note: Characterising and comparing different palaeoclimates with dynamical systems theory" by Messori and Faranda

Recommendation: Major Revisions

This manuscript promotes dynamical systems measures for the evaluation and comparison of climate model simulations and climate data sets in general. While dynamical systems methods can surely provide additional insight in climate data, the present study does not make a very convincing case. I do not see the advantage over some other used metrics. So my recommendation is for major revisions before the manuscript can be considered for publication.

[Figure]

1) Looking at Fig. 1 I do not see how to gain additional insight from the two dynamical systems measures compared with inspecting the precipitation (Fig. 2c).

Only for the control simulation uncertainty bounds are given. My guess is that uncertainty bounds for the other two simulations would overlap with the bounds of the control simulation. If this is the case then one cannot say that the measures actually show any significant differences. They are already now almost always in the uncertainty bounds of the control simulation. So, what do we really learn from this?

2) I do not think that the typical Climate of the Past reader is very familiar with dynamical systems concepts like Poincare recurrences, Axiom A system, etc. The authors should explain them more carefully and in an intuitive way. Has it actually been shown that the climate system is Axiom A?

Perhaps the authors should first provide an intuitive introduction to the concepts and methods and move the more technical details to an appendix.

―――――――――――――――――――――

---

## Referee Comment (RC2) · Anonymous Referee #2 · 14 Sep 2020

In this technical note, the authors introduce their method of recurrence-based "dynamical systems analysis" to the field of Paleoclimate simulations. By estimating the statistics of extremely close recurrences in one or two variables, their method summarizes the instantaneous state of a dynamical system by the effective degrees of freedom, the persistence, and the coupling between different variables. As an example application, the authors show how their metrics can detect differences in the dynamics of Mid-Holocene North African Monsoon circulation under various vegetation and aerosol forcings.

Overall, I would agree that the approach presented here can be especially helpful for

the analysis of Paleo-simulations: Uncertain forcing and boundary conditions can potentially lead the simulated system into unknown dynamical regimes, which might differ from our present-day intuitions in unexpected ways. Objective, generally applicable measures of the overall dynamical behavior may be rather helpful in detecting differences and similarities of various simulated climates.

The manuscript is very well-written and overall fits the scope of the journal. The description of the method and the presented case study do, however, raise several concerns which should be addressed before publication.

**Main points**: The assumption of stationarity is not sufficiently discussed. If a substantial regime shift occurs at some point during the simulation, I would expect that recurrences will only be observed within each regime but not between the two. Doesn't the estimated dimension then depend massively on the length of the time-series before and after the shift? Say we have simulated 1000 years before and 9000 years after a de-glaciation phase. Won't the dimension in the first 1000 years be artificially increased just because there are fewer recurrence candidates? Wouldn't this result change completely if we had stopped the simulation 1000 years after the shift?

The use of binary precipitation fields seems worrying: If I understand correctly, the distance measure is then effectively no longer a continuous but a discrete random variable. Do we know that the theoretical limit results apply in the discrete case?

Depending on the domain, we might have many time-steps with identical zero precipitation. What happens to theta and d in such cases? Shouldn't the persistence be effectively infinite and the dimension zero if two subsequent time steps are exactly identical?

Please include significance tests for your composites (Fig.3, 5, 6). As it is, we don't know which of these patterns might just be random chance.

**Minor points**: Neither the abstract not the Motivation chapter gives the reader any

idea what the dynamical indicators actually do. "Different dynamical properties" (l.9) is too vague. Please add at least an intuitive explanation of what kinds of properties you mean.

L.20 "new challenges" I'm no expert on this but you cite Paleo-simualtions going back at least to 1996, this is hardly a "new" issue.

L.30-33 this paragraph is copied nearly verbatim from the abstract, maybe instead you could give some more explanation of what the dynamical indicators actually do.

You mention that theta and alpha are bounded, what about d?

Section 2.2: Maybe mention that x(t) corresponds to the sea level pressure or precipitation field from the example before.

Eq.3 looks like alpha was asymmetric with respect to x and y because the denominator contains only x, but nu( g(x) > sx ) = nu( g(y) > sy ) = 1 - q, correct ? Maybe make that more clear.

L. 155 how do you arrive at these definitions of Monsoon and Pre-Monsoon? Are those the present day conditions?

Fig.1 there is almost no visible difference between b and c, maybe plot the difference between GS-PD / GS-RD and CNTL instead?

Fig. 2 Please explain more specifically what a) and b) are telling us besides the shift in monsoon onset and ending, both of which we already see in c). In particular, how do you interpret the fact that the maxima in d shift from blue to red, but the decrease in theta (b) happens nearly at the same time in all three curves.

Also in Fig.2 there is no appreciable difference between the red and yellow curve. Do these systems have different dynamics or not?

In Fig.5 a) and b) (also Fig. 6) it is impossible to tell what the actual values of the contours are because there are only negative anomalies. Maybe add labels? In any

case this figure in particular needs a significance test in order to decide which patterns are actually worth interpreting.

Section 4: You say that your method can complement "other", "conventional" approaches but never name any of these other techniques. Can you give an example of a standard method with similar goals as yours? Perhaps PCA? That might help readers grasp what (approximately) your method does. You could also discuss some similarities or differences, highlighting what sets your approach apart.

L. 285 can you please be a little more specific than "several good recurrences"? Very roughly how much data do I need for this method? How can I check if I have sufficiently "good" recurrences?

―――――――――――――――――――――――

---

## Author Comment (AC1) · 13 Nov 2020

**Replies to Reviewer #1**

November 2020

We would like to thank the Reviewer for the constructive set of comments provided. We provide a detailed reply (in red) to the individual comments (in *italics*) below.

*While dynamical systems methods can surely provide additional insight in climate data, the present study does not make a very convincing case. I do not see the advantage over some other used metrics.*

We take the Reviewer's concern very seriously, as our main aim in this technical note was not to present a full scientific analysis, but rather to motivate the usefulness of concepts from dynamical systems theory in the analysis of large palaeoclimate simulations. We detail some changes we have implemented in our study to better motivate the use of the dynamical systems approach, in our reply the first part of the Reviewer's Comment #1. Partly in response to one of the comments by Reviewer #2, we have further added to Sect. 4 a discussion of the merits of our approach relative to other commonly used statistical methods. We specifically consider the widely used weather regimes (which are often derived from PCA or clustering approaches) and the Canonical Correlation Analysis, which identifies linear combinations of two variables with maximum correlation. Finally, always in Sect. 4, we have included a discussion of the possible applications of our approach to the identification of tipping points in climate datasets and to other palaeoclimatic problems, such as diagnosing the responses of different numerical models to a given forcing.

1.

– *Looking at Fig. 1 I do not see how to gain additional insight from the two dynamical systems measures compared with inspecting the precipitation (Fig. 2c).*

– *Only for the control simulation uncertainty bounds are given. My guess is that uncertainty bounds for the other two simulations would overlap with the bounds of the control simulation. If this is the case then one cannot say that the measures actually show any significant differences. They are already now almost always in the uncertainty bounds of the control simulation. So, what do we really learn from this?*

Concerning the Reviewer's first point we assume s/he was referring to Figs. 2a, b versus Fig. 2c in the original draft. Indeed, Fig. 1 in the original draft is not related to dynamical systems, and was simply provided to illustrate the climatology of the simulations we are analysing. Focusing the discussion on Fig. 2 in the original draft (Fig. 3 in the revised manuscript), we argue that the dynamical systems metrics provide a very efficient tool to rapidly identify salient differences between the simulations and enable to put

forth mechanistic hypotheses on their origins. For example, neither Fig. 1 nor Fig. 2c give any indication as to the mechanisms driving monsoonal precipitation in the three simulations. The increased persistence in the $\text{MH}_{GS+PD}$ and $\text{MH}_{GS+RD}$ simulations, shown in Fig. 2b, immediately points to the fact that the role of transient atmospheric features is likely weakened compared to the $\text{MH}_{CNTL}$ run. This hypothesis would then need to be verified with detailed analyses of atmospheric dynamics, but the computationally inexpensive $\theta$ metric is nonetheless valuable in pointing to it as an interesting aspect to investigate further. Similarly, having ascertained that $d$ is sensitive to the monsoon's onset, its interannual variability can be used to quantify the variability of the monsoon's onset within each model simulation. If one were using more conventional analysis techniques, this would likely require defining and computing a monsoon onset index. It is further an aspect which could be easily overlooked, seeing as an inspection of a simple seasonal cycle of precipitation does not evidence the pre-monsoon season as being particularly variable, while for $d$ it is the season displaying the largest variability (see blue shading in old Fig. 2a, c, now Fig. 3a, c). Although we appreciate that this goes beyond the part of the analysis the Reviewer was commenting on, we would like to highlight that the metric which is perhaps the most valuable in providing additional insights compared to more conventional analyses is $\alpha$. It is not easy to design a computationally efficient, multivariate statistical dependence measure providing a value for each timestep of a dataset. As we show, $\alpha$ can help to probe in a 1-D space the role of large-scale atmospheric circulation anomalies in favouring the northward extension of the monsoonal precipitation. As stated above, we take the Reviewer's concerns on the usefulness of the approach we propose very seriously, since they point to the fact that we have not illustrated the above aspects clearly in our text. We have therefore updated the initial part of Sect. 3.2 and parts of Sect. 4 to reflect more explicitly the value that we believe lies in the application of the dynamical systems metrics we propose.

Concerning the Reviewer's second point, we agree that we should have placed greater care in some statistical aspects of our analysis. We provided an indication of the standard deviation of the control simulation in the figure as reference for the metrics' variability. However, we did not mean to use it as a statistical test of whether the curves are different or not. Indeed, we are not interested in testing the difference on single days (which is what such standard deviation bounds would be showing), but rather on testing whether our approach detects a significant difference in the atmospheric/monsoonal dynamics between the simulations, i.e. a difference over the whole monsoon season or year. To this effect, we have performed a Wilcoxon rank sum test, which enables us to make a statement as to whether the data from two samples is drawn from continuous distributions with equal medians or not. We performed the test at the 1% level to compare the medians of the control simulation to the two perturbed simulations, for the dynamical systems metrics shown in Figs. 2a, b, and 4a, b (Figs. 3 and 5 in the revised manuscript) for both the Pre-Monsoon Season, Monsoon Season and the whole year. We found that in all cases the null hypothesis of the population medians being equal is rejected. Based on this test, we conclude that the dynamical systems metrics highlight a significant difference between the control and perturbed simulations under both the Pre-Monsoon and Monsoon seasons. We have now added a description of these results to Sect. 3.2 in our manuscript, focusing for conciseness on the monsoon season. We further discuss the large variability within each run which, although it does not preclude the statistical significance of our results, is nonetheless a relevant aspect to touch upon.

*2. I do not think that the typical Climate of the Past reader is very familiar with dynamical systems concepts like Poincaré recurrences, Axiom A system, etc. The authors should explain them more carefully and in an intuitive way. Has it actually been shown that the climate system is Axiom A? Perhaps the authors should first provide an intuitive introduction to the concepts and methods and move the more*

*technical details to an Appendix.*

We understand the importance of making our methodology accessible to a readership which may not be familiar with jargon specific to the dynamical systems community. We have now expanded Sect. 2.1 to improve the qualitative explanation of the methodology. We have specifically added analogies to the flow of raindrops on topography as intuitive "mental anchors" that may be related directly to the concepts of local dimension, persistence and co-recurrence. At the same time, since this is a technical note, we would like to keep the more formal description of the methodology in the main text (Sect. 2.2). We have, however, removed some non-essential jargon (including the term "Axiom A") and added explanations of some uncommon terms to the section, so that a technically-minded reader may be able to follow the derivation without needing to have a background in dynamical systems theory. We have further added a new Fig. 1 to this section to provide a graphical illustration of our approach, and thus facilitate the understanding of the metrics' computation. To clarify this structure, we have added a short introductory paragraph to Sect. 2. Concerning Axiom A systems, (Eckmann and Ruelle, 1985; Ruelle, 2009) we recall that these are a special class of dynamical systems possessing a Sinai-Ruelle-Bowen (SRB) invariant measure (Young, 2002) and featuring uniform hyperbolicity in the attracting set. Such invariant measure is robust against infinitesimal stochastic perturbations, namely it coincides with the Kolmogorov physical measure. Another important property of Axiom A systems is that it is possible to develop a response theory for computing the change in the statistical properties of any observable due to small perturbations in the flow (Ruelle, 1976, 2009). While they may seem a purely theoretical construct, Axiom A systems do have a direct relevance for geophysical fluid dynamics applications, and climate models themselves behave much like Axiom A systems (Ragone et al., 2016). A very nice explanation of the relevance of Axiom A systems for climate science can be found in Lucarini and Bodai (2017): "Axiom A systems are indeed far from being typical dynamical systems, but, according to the chaotic hypothesis of Gallavotti and Cohen (1995), they can be taken as effective models for chaotic physical systems with many degrees of freedom. Specifically, this means that when looking at macroscopic observables in sufficiently chaotic (to be intended in a qualitative sense) high-dimensional systems, it is extremely hard to distinguish their properties from those of an Axiom A system, including some degree of structural stability. One can interpret the chaotic hypothesis as the possibility of constructing robust physical properties for the system under investigation. Therefore, providing results for Axiom A systems can be thought of as being of rather general physical relevance." As mentioned above, we have chosen to remove the term "Axiom A" from the paper, as in earlier works (e.g. Faranda et al. 2017), we have seen that the dynamical systems metrics can be computed for non-uniformly hyperbolic attractors, providing insights in the dynamics such as the existence and the properties of singular points. In the new version of the manuscript, we instead explicitly list the theoretical requirements for a compact attractor and a stationary system, which we found can affect the computation of the dynamical metrics (see, e.g. Faranda et al. 2019b). However, as the Reviewer comments and author replies remain public in the CP discussion, we deemed it important to address the Reviewer's very pertinent comment in some detail here.

As a final note, we would like to highlight two changes we have implemented in the paper beyond those described in the replies to the Reviewer. The first is that we have decided to remove Fig. A3. This was the only figure showing year-round geographical anomalies. It was barely mentioned in the text, since our analysis of the geographical anomalies focuses on the rainy season, and upon reviewing the manuscript we did not think it contributed with meaningful information to the overall discussion. The second change is that, in response to comment #3 by Reviewer #2, we have decided to exclude some datapoints from the

analysis based on theoretical considerations on the computation of the dynamical systems metrics. This has led to minor changes in some of the figures, but does not alter any of our qualitative conclusions.

**Additional References not Cited in the Study**

- Eckmann, J-P., and David Ruelle. "Ergodic theory of chaos and strange attractors." The theory of chaotic attractors. Springer, New York, NY, 1985. 273-312.

- Gallavotti, Giovanni, and Ezechiel Godert David Cohen. "Dynamical ensembles in nonequilibrium statistical mechanics." Physical review letters 74.14 (1995): 2694.

- Lucarini, Valerio, and Tamás Bódai. "Edge states in the climate system: exploring global instabilities and critical transitions." Nonlinearity 30.7 (2017): R32.

- Ragone, Francesco, Valerio Lucarini, and Frank Lunkeit. "A new framework for climate sensitivity and prediction: a modelling perspective." Climate Dynamics 46.5-6 (2016): 1459-1471.

- Ruelle, David. "A measure associated with axiom-A attractors." American Journal of Mathematics (1976): 619-654.

- Ruelle, David. "A review of linear response theory for general differentiable dynamical systems." Nonlinearity 22.4 (2009): 855.

- Young, Lai-Sang. "What are SRB measures, and which dynamical systems have them?." Journal of Statistical Physics 108.5-6 (2002): 733-754.

---

## Author Comment (AC2) · 13 Nov 2020

**Replies to Reviewer #2**

November 2020

We would like to thank the Reviewer for the constructive set of comments provided. We provide a detailed reply (in red) to the individual comments (in *italics*) below.

**Main points**

1. *The assumption of stationarity is not sufficiently discussed. If a substantial regime shift occurs at some point during the simulation, I would expect that recurrences will only be observed within each regime but not between the two. Doesn't the estimated dimension then depend massively on the length of the time-series before and after the shift? Say we have simulated 1000 years before and 9000 years after a deglaciation phase. Won't the dimension in the first 1000 years be artificially increased just because there are fewer recurrence candidates? Wouldn't this result change completely if we had stopped the simulation 1000 years after the shift?*

The Reviewer raises a very good point about data stationarity. However, we believe that the Reviewer's comment combines two distinct issues. The first is how the length of the dataset affects the local dimension; the second concerns regime shifts in the data. The local dimensions depend on the number of "good" recurrences in the dataset chosen, where the definition of "good" will depend on the data and distance metric dist being used. Indeed a high dimension may correspond to few good recurrences. While very short data series do not allow to identify good recurrences, and hence valid estimates of the local dimension, once a "reasonably long" timeseries is obtained, our estimate of the local dimension is only weakly affected by the number of available years of observations. The meaning of "reasonably long" is hard to define formally, but we found empirically that several decades of daily data usually fit this definition when studying atmospheric data at synoptic or larger scales. Indeed, previous studies have outlined that the convergence of our estimators is relatively fast for daily atmospheric data in quasi-stationary conditions. For example, in Faranda et al. (2017b), we have analysed both the ERA-Interim (1979-2015) and NCEP/NCAR (1948-2015) reanalyses and found that they provide very similar values of local dimension and persistence, despite the different length of the time series. In Buschow and Friedrichs (2018), the authors analysed a 1000 year-long simulation from the simplified climate model PlaSim. Using daily data in stationary conditions they found that, as they increased the data used from a fraction of the total dataset up to the full 1000 years, the values of the dimension drifted very slowly.

Concerning the issue of non-stationarity, in Faranda et al. (2019a) we showed that our methodology is able to detect weak non-stationarities in the climate system, as for example is the case for the ongoing climate change. An abrupt regime shift poses a different challenge, and what is the limit of validity of our

metrics for non-stationary systems is very much an open question. Our educated guess for the case put forth by the Reviewer is that, as long as each "regime" is sufficiently long to provide a reasonable number of "good" recurrences within it, then differences in the length of the different regimes may not be a deal-breaker for reliable estimates of d. Although we did not formally test this hypothesis on palaeoclimatic data showing bifurcations, we have indeed analysed the transitions between different jet regimes in both a conceptual coupled lattice map (Faranda et al., 2019) and a quasi-geostrophic model (Messori et al., in review). The results show that different values of $d$ and $\theta$ successfully reflect these transitions, and thus may be interpreted as associated with different basins of attraction of the system. However, we currently have no formal arguments to support this observation. Clearly, if one is interested in mean dimension over the whole simulation, then the relative length of the pre- and post-shift simulated timeperiods will play a major role. However, upon inspection of a timeseries of $d$ it would be natural to compute separate statistics before and after the regime shift (and hence shift in $d$). We now comment on the above points, which we recognise can be of relevance to a number of palaeoclimatic applications, in Sect. 4 of the revised text.

2. *The use of binary precipitation fields seems worrying: if I understand correctly, the distance measure is then effectively no longer a continuous but a discrete random variable. Do we know that the theoretical limit results apply in the discrete case?*

Precipitation data is a somewhat peculiar case since they have a fractal structure rather than that of a discrete variable (see e.g. Lovejoy and Schertzer, 1985). This holds even after binary discretisation is applied. The prp variable we analyse in our study is thus effectively a fractal spatial set (Langousis, 2009; Brunsell, 2010). Lucarini et al. (2012), have addressed the question of recurrences for fractal sets using iterated function systems which generate Cantor sets. Fractal data still have as limiting continuous model a generalized extreme value distribution, and therefore the estimates of the dimension and the persistence are possible for this kind of datasets. Indeed, in Faranda et al. (2017a), we have appplied the recurrence method to precipitation data extracted by the NCEP datasets and shown that the results have a physically meaningful interpretation. We thus have robust theoretical arguments supporting the use of our approach on discretised precipitation data. We have now added a short mention of this point in the text, when introducing the prp variable in Sect. 3.1.

3. *Depending on the domain, we might have many time-steps with identical zero precipitation. What happens to theta and d in such cases? Shouldn't the persistence be effectively infinite and the dimension zero if two subsequent time steps are exactly identical?*

The Reviewer raises a very good point. We had initially decided not to discuss this in the paper as we deemed it a technical subtlety, but we now realise that this omission may cause difficulties to those readers who may wish to apply the methodology we propose. If there are two timesteps in the dataset which are identical, this leads to the "distance" between the state of interest and the recurrence being 0. In practice, this would not preclude identifying a finite threshold to identify recurrences and then applying our algorithm. However, from a theoretical standpoint, in the limit of an infinitely long timeseries the existence of identical states in the chosen variable underscores a $d = 0$ for those states. In our calculations, we therefore assign $d = 0$. Since these states have the dimension of a point and do not reflect any dynamical information, we have decided to exclude them from our calculations in the revised study. A second, related, issue is that all the "good" recurrences may be identical. This is for example the case for a day with very

little precipitation, whose closest recurrences are all the days without precipitation. In this case it is not possible to compute a meaningful recurrence threshold and we again discard the datapoint. Our algorithm for computing $\theta$ is also affected by these issues, and in order to obtain valid estimates of persistence one should revert to a naïve calculation of the average number of consecutive identical timesteps. In our analysis, we have decided to discard these states, for consistency with the calculation of $d$. This choice has led to minor changes in some of the figures, but does not alter any of our qualitative conclusions. Concerning the final part of the Reviewer's comment, $\theta$ can only be zero at a fixed point of the system, i.e. if all successive timesteps bring no change to the state of the system. A trivial example is a pendulum in its equilibrium position (or the equilibrium climate of a hypothetical planet at 0 K without any external energy input). As such, having two (or more) successive timesteps which are identical does not imply infinite persistence. We have now included a brief discussion of these technical yet very important points in Sect. 2.2.

*4. Please include significance tests for your composites (Fig. 3, 5, 6). As it is, we don't know which of these patterns might just be random chance.*

We fully agree with the Reviewer on this point. We have now computed the one-sided 5% significance bounds for the positive and negative precipitation anomalies respectively, in the three figures in the main text (and the corresponding figures in the Appendix) by bootstrap resampling with 1000 iterations. The results of this test are now shown in the figures. We have also updated the text to include a discussion of the significance of the anomalies when referring to the figures.

**Minor points**

*5. Neither the abstract not the Motivation chapter gives the reader any idea what the dynamical indicators actually do. "Different dynamical properties" (l.9) is too vague. Please add at least an intuitive explanation of what kinds of properties you mean.*

We have added a short qualitative interpretation of the three metrics to the introductory section. In the abstract, we have added some dynamical systems keywords, such as "persistence" and "instantaneous", which were previously missing and may help the readers form a better picture of the contents of the paper. We then expand upon these terms in the main text.

*6. L.20 "new challenges" I'm no expert on this but you cite Paleo-simualtions going back at least to 1996, this is hardly a "new" issue.*

We understand the apparent contradiction. We have now clarified in the introduction that the challenge arises from the exponential increase in the amount of data generated by numerical simulations of (palaeo)climates, which has gone hand in hand with the development of ever more complex and highy-resolved models. The study we cite from 1996 is indeed one of the earliest examples of numerical simulations for the MH Green Sahara episode, and likely produced an amount of data which is orders of magnitudes smaller than the data produced by a modern numerical climate model, such as the one used to perform the simulations analysed here.

7. *L.30-33 this paragraph is copied nearly verbatim from the abstract, maybe instead you could give some more explanation of what the dynamical indicators actually do. You mention that theta and alpha are bounded, what about d?*

As part of the new qualitative interpretation of the three metrics we added to the introductory section (see our reply to comment #5 above), we have rephrased this paragraph. Concerning the bounds of the indicators, $d$ is a positive real number. Technically $0 \leq d \leq +\infty$ with $d = 0$ being the dimension of a point and $d = +\infty$ being the dimension of an unbounded infinite dimensional system with no attractor (e.g. a brownian motion in infinite dimensions). Moving to the persistence, $0 \leq \theta \leq 1$. $\theta = 0$ is the limiting case of a fixed point and $\theta = 1$ the case of points immediately leaving the neighborhood of $\zeta$. Finally $0 \leq \alpha \leq 1$ with $\alpha = 0$ being the case of non co-recurring variables and $\alpha = 1$ the case of perfect synchronization. We now state these bounds and their interpretation in Sect. 2.2.

8. *Section 2.2: Maybe mention that x(t) corresponds to the sea level pressure or precipitation field from the example before.*

We have added this, as suggested by the Reviewer.

9. *Eq.3 looks like alpha was asymmetric with respect to x and y because the denominator contains only x, but nu( g(x) ≥ sx ) = nu( g(y) ≥ sy ) = 1 - q, correct ? Maybe make that more clear.*

Since excedances in both x and y are defined relative to the same high quantile, the Reviewer is absolutely correct. We have clarified this point in the text in Sect. 2.2.

10. *L. 155 how do you arrive at these definitions of Monsoon and Pre-Monsoon? Are those the present day conditions?*

These definitions are indeed taken from the present-day climatology of the West African Monsoon. We use them as reference periods to highlight the changes in the timing of the monsoonal onset and decay in the Green Sahara or Green Sahara and Reduced Dust simulations. We now specify this in the text.

11. *Fig.1 there is almost no visible difference between b and c, maybe plot the difference between GS-PD / GS-RD and CNTL instead?*

We have now added a new panel (d) to the figure which, as suggested by the Reviewer, shows the difference in precipitation between the two Green Sahara simulations. We briefly comment on this in the revised text. We have also implemented a corresponding change in Fig. A3 (Fig. A4 in the original submission).

12. *Fig. 2 Please explain more specifically what a) and b) are telling us besides the shift in monsoon onset and ending, both of which we already see in c). In particular, how do you interpret the fact that the maxima in d shift from blue to red, but the decrease in theta (b) happens nearly at the same time in all three curves.*

We argue that the three panels of Figure 3 (Figure 2 in the original submission) provide largely complementary information. For example, Fig. 3c does not give any indication as to the mechanisms driving monsoonal precipitation in the three simulations. The increased persistence in the $MH_{GS+PD}$ and $MH_{GS+RD}$ simulations, shown in Fig. 3b, immediately points to the fact that the role of transient atmospheric features is likely weakened compared to the $MH_{CNTL}$ run. This hypothesis would then need to be verified with detailed analyses of atmospheric dynamics, but the computationally inexpensive $\theta$ metric is nonetheless valuable in pointing to it as an interesting aspect to investigate further. Similarly, having ascertained that $d$ is sensitive to the monsoon's onset, its interannual variability can be used to quantify the variability of the monsoon's onset within each model simulations. If one were using more conventional analysis techniques, this would likely require defining and computing a monsoon onset index. It is further an aspect which could be easily overlooked, seeing as an inspection of a simple seasonal cycle of precipitation does not evidence the pre-monsoon season as being particularly variable, while for $d$ it is the season displaying the largest variability (see blue shading in Fig. 3a, c). Also in response to comment #1 by Reviewer #1, we have updated the initial part of Sect. 3.2 to reflect more explicitly the information provided by the first two panels of Fig. 3. Concerning the last part of the Reviewer's comment, we would argue that there is a large shift also in the decrease in $\theta$ across the three simulations. Indeed, looking at Fig. 3c, the blue curve shows a drop between days $\sim$100 to 135, and a rise between days $\sim$275 to 300. The orange and red curves display a drop between days $\sim$60 to 100, and a rise between days $\sim$290 to 315. The shift in timing between the blue curve on the one hand and the orange and red curves on the other hand, is comparable to the shift seen in panel (a) for the increase/decrease in $d$ at the onset/end of the monsoon season. The maximum $d$ for the blue curve is achieved in the early monsoon season, at a time when the other two curves show a local (albeit not absolute) maximum. We therefore do not see any inconsistency between the relative changes in $d$ and $\theta$ curves across the three simulations.

13. *Also in Fig. 2 there is no appreciable difference between the red and yellow curve. Do these systems have different dynamics or not?*

Previous analysis of these same simulations (e.g. Gateani et al., 2017) and studies from other authors (e.g. Thompson et al., 2019) suggest that, compared to the effect of Saharan Greening, the dust reduction under a Green Sahara scenario only has limited impacts on the atmospheric circulation. Moreover, the same studies have shown that the changes in atmospheric (thermo)dynamics leading to increased monsoonal precipitation in the $MH_{GS+PD}$ and $MH_{GS+RD}$ simulations are similar in nature. Indeed, the type of changes seen between the $MH_{CNTL}$ and $MH_{GS+PD}$ and $MH_{CNTL}$ and $MH_{GS+RD}$ simulations are very similar. We now discuss this briefly in section 4, highlighting that this shows how our approach may not be indicated for cases where the analysis focuses on similar climates displaying comparable (thermo)dynamical properties.

14. *In Fig.5 a) and b) (also Fig. 6) it is impossible to tell what the actual values of the contours are because there are only negative anomalies. Maybe add labels? In any case this figure in particular needs a significance test in order to decide which patterns are actually worth interpreting.*

Concerning the contours, the Reviewer is entirely right. We now specify the range of SLP and Z500 anomalies shown in each of the panels in the figure captions. We have opted not to add labels directly in the figures as these either covered the precipitation anomalies (if on a white background) or were not easily legible (if on a transparent background). Regarding significance, following Major Comment #4, we

have added significance bounds on the precipitation anomalies. We focus on these since we are interested in detecting significant changes in monsoonal precipitation and then being able to relate these to specific SLP and Z500 patterns. The interest in the latter is not to have a locally significant anomaly in terms of magnitude (which is what standard significance tests verify), but rather a spatially coherent pattern which can explain the significant precipitation anomalies.

15. *Section 4: You say that your method can complement "other", "conventional" approaches but never name any of these other techniques. Can you give an example of a standard method with similar goals as yours? Perhaps PCA? That might help readers grasp what (approximately) your method does. You could also discuss some similarities or differences, highlighting what sets your approach apart.*

We have added a paragraph on this issue in Sect. 4, as suggested by the Reviewer. In previous work, we have compared: (i) the information extracted with $d$ and $\theta$ to that obtained via the analysis of weather regimes (e.g. Faranda et al., 2017b; Hochman et al., 2019), often obtained by PCA or clustering methods; and (ii) the results obtained from $\alpha$ with those extracted from Canonical Correlation Analysis (De Luca et al., 2020b). Concerning (i), we found that the dynamical indicators provide an (almost) continuous counterpart to the heavily discretised PCA/weather regime description of atmospheric variability. We specifically found that, for a North Atlantic domain, compositing days falling in different quadrants of the $d$–$\theta$ space allows to recover the four canonical weather regimes (Faranda et al., 2017b). At the same time, the dynamical systems metrics provide a wealth of additional information on how the atmosphere moves within and between these weather regimes. Concerning (ii), we found that the information derived from $\alpha$ largely overlaps that of the CCA, yet that $\alpha$ was able to better capture the footprint of co-variability for extreme events (De Luca et al., 2020b). Although not applied in this specific study, the definition of $\alpha$ can be easily extended to a multivariate case beyond two variables, while the CCA framework requires more complex adaptations (such as partial CCA). Finally, we remark that while statistical techniques can provide valuable information on the correlation structures of recurrences, the dynamical indicators are rooted in the causal structure of the underlying dynamics of the system (e.g. a low dimension does not just point to a specific metastable state of the dynamics – or a principal component – but also informs that this state is in a more predictable region of the attractor).

16. *L. 285 can you please be a little more specific than "several good recurrences"? Very roughly how much data do I need for this method? How can I check if I have sufficiently "good" recurrences?*

While we do not have a definitive theoretical answer to this question, we have devised some procedures to test the recurrence statistics. In Faranda et al. (2011), when first applying the technique to numerical data issued from dynamical systems, we verified via a Lilliefors test whether the distribution achieved by recurrences after application of the logarithmic weight is a Gumbel law. Since deviations from this law are observed at finite time for all statistical datasets (see, e.g. Gomes and De Haan, 1999), in more recent studies (e.g. Faranda et al., 2017b) we tend to prefer sensitivity tests for the obtained values, for example by reducing the length of the datasets and repeating the estimates to check their stability. This is also the strategy chosen by Buschow and Friedrichs (2018). We now included a brief description of these issues in Sect. 4

As a final note, we would like to highlight a further change that we have implemented in the paper beyond those outlined in the replies to the Reviewer. Specifically, we have decided to remove Fig. A3. This was the only figure showing year-round geographical anomalies. It was barely mentioned in the text, since our analysis of the geographical anomalies focuses on the rainy season, and upon reviewing the manuscript we did not think it contributed with meaningful information to the overall discussion.

**Additional References not Cited in the Study**

- Gomes, M. Ivette, and Laurens De Haan. "Approximation by penultimate extreme value distributions." Extremes 2.1 (1999): 71-85.

---

## Author Response (AR1)

Dear Reviewers,

We have revised our manuscript as per the replies we provided in the public discussion. Below, we enclose a version of the new manuscript with edits marked in red. We trust that these reflect your input and we look forward to receiving your feedback on the revised text.

Best Regards,

Gabriele Messori and Davide Faranda

[revised manuscript text omitted]

---

## Author Response (AR2)

**Replies to Reviewer #2**

January 2021

We would like to thank the Reviewer for carefully evaluating our revised manuscript, and for their suggestions for further improvement. We provide a detailed reply (in red) to the individual comments (in *italics*) below.

**Main points**

1. *Section 2.1 I overall approve of the "raindrop" analogy although it does become a little strained when the vegetation is added. Unfortunately I don't have a better idea either. Maybe it would have been more consistent to show a stylized image of a raindrop flowing down a rock instead of the Lorenz attractor in Figure 1. The zoomed in parts of the trajectories could easily represent paths of rain drops instead of the abstract trajectory along the "butterfly". On the other hand most readers with any kind of climate science background are probably familiar with L63.*

We understand the Reviewer's viewpoint, and have opted to add a figure (new Fig. A1) showing a schematic of raindrops flowing down the side of a valley. We hope that this makes the raindrop analogy (including the vegetation component) clearer, while also leaving the familiar reference of the Lorenz '63 attractor for the readers with a background in dynamical meteorology.

2. *Answer to reviewer #2 major comment 2: I would like to clarify my original comment because I don't think it came across properly. If the prp field contains only ones and zeros, the distance between such fields is effectively an integer. Regardless of the regular or fractal nature of the binary point set, the distance d is a fundamentally different kind of random variable than for slp and z500 because its density consists of a finite number of point masses at whole numbers. The members of the GEV family, on the other hand, are continuous functions. I appreciate that your methodology may nonetheless be applicable but I do not see the "robust theoretical arguments": In Lucarini et al. (2012), the attractor of the system is a fractal but the considered distance measure is continuous and not discrete. Faranda et al 2017a present an empirical study which may give evidence that this strategy for precipitation differences is appropriate in practice but they do not give theoretical arguments for its validity. I therefore believe that intermittent variables like precipitation are not ideal targets for your kind of analysis.*

We thank the Reviewer for this clarification, as we had indeed partly misunderstood their original comment. Our estimation procedure is based on fitting the exponential member of the GPD family in the context of dynamical systems theory. Hitz (2016) showed that discrete variables which yield the same mixing properties of continuous variables can still be approximated by GPD distributions. Here, we

implicitly use this result which, as shown by Hitz, may be applied to modelling geophysical data. In the new version of the manuscript, we have expanded our discussion of this theoretical, yet very important point. In view of the clarification provided by the Reviewer, we have also revised our previous statement and now write that there is no complete theoretical framework for the application of extreme value theory to recurrences of discrete fields, although studies such as Hitz (2016) and Faranda *et al.* (2017a) support the physical relevance of the results .

3. *L.136, equation (3): What do you actually mean by the "U"-symbol? I thought this was the symbol for the union of two sets but the expressions left and right of it are not sets but logical statements in which case you probably mean "v" for a logical "or"? In either case I think what you actually want is a logical "and" which would be an inverted "v" (in Faranda et al. 2020 it is formulated in terms of conditional probability using the "|" symbol which also means that both thresholds must be exceeded): If you actually meant "or", the numerator would be the number of cases where either g(x) or g(y) exceed their thresholds which is at least as large as the number of cases where only g(x) is above $s_x$; alpha would then be bounded from below by 1.*

We thank the Reviewer for spotting this typo; we should indeed have used the "|" symbol, and have now corrected this in the paper.

4. *L. 306. "significantly increased precipitation across the southern portion of the domain [are] favored by negative SLP and Z500 anomalies to the North of the strongest precipitation anomalies [...]. These are likely the footprint of a strengthened heat low [...]" Heat lows are generally shallow structures with negative pressure anomalies in the lower levels and divergence with associated positive pressure (or geopotential) anomalies above. In their discussion of West African heat lows, Lavaysse et al. (2009) explicitly state that "A heat low is an area of low atmospheric pressure near the surface resulting from heating of the lower troposphere and the subsequent lifting of isobaric surfaces and divergence of air aloft." If anything, I would expect a positive geopotential anomaly at 500hPa associated with this kind of pressure system. Your interpretation of the negative Z500 anomaly as an indicator of increased heat low activity is therefore questionable.*

The Reviewer is absolutely correct; we had mistakenly written "negative [...] Z500 anomalies" instead of "positive [...] Z500 anomalies." Indeed, Fig. 7a clearly shows negative Z500 anomalies in the south of the domain, and positive Z500 anomalies to the North of the negative SLP anomaly core in Fig. 6a, much like what is shown in Fig. 2b in Lavaysse et al. (2009). We have now rephrased this passage to: "In $MH_{CNTL}$, this takes the form of significantly increased precipitation across the southern portion of the domain, favoured by negative SLP anomalies to the North of the strongest precipitation anomalies (Fig. 6a) and positive Z500 anomalies to the North of the negative SLP core (Fig. 7a). These are likely the footprint of a strengthened heat low (see e.g. Fig. 2b in Lavaysse *et al.*, 2009), which favours a northward progression of the monsoonal precipitation." We have further reformulated some of the sentences in the rest of the paragraph to ensure a more accurate description of the observed Z500 anomalies.

**References**

Hitz, Adrien. Modelling of extremes. 2016. PhD Thesis. University of Oxford.